# Pervasive Transcription-coupled DNA repair in *E. coli*

Britney Martinez[1], Binod K. Bharati[1,2], Vitaly Epshtein[1] & Evgeny Nudler [1,2 ✉]

Global Genomic Repair (GGR) and Transcription-Coupled Repair (TCR) have been viewed, respectively, as major and minor sub-pathways of the nucleotide excision repair (NER) process that removes bulky lesions from the genome. Here we applied a next generation sequencing assay, CPD-seq, in *E. coli* to measure the levels of cyclobutane pyrimidine dimer (CPD) lesions before, during, and after UV-induced genotoxic stress, and, therefore, to determine the rate of genomic recovery by NER at a single nucleotide resolution. We find that active transcription is necessary for the repair of not only the template strand (TS), but also the non-template strand (NTS), and that the bulk of TCR is independent of Mfd – a DNA translocase that is thought to be necessary and sufficient for TCR in bacteria. We further show that repair of both TS and NTS is enhanced by increased readthrough past Rho-dependent terminators. We demonstrate that UV-induced genotoxic stress promotes global antitermination so that TCR is more accessible to the antisense, intergenic, and other low transcribed regions. Overall, our data suggest that GGR and TCR are essentially the same process required for complete repair of the bacterial genome.

[1] Department of Biochemistry and Molecular Pharmacology, NYU Grossman School of Medicine, New York, NY 10016, USA. [2] Howard Hughes Medical Institute, NYU Grossman School of Medicine, New York, New York 10016, USA. ✉email: evgeny.nudler@nyulangone.org

Nucleotide excision repair (NER) is a highly evolutionary conserved process that is required for the removal of bulky DNA lesions, such as those caused by UV light[1]. In bacteria, the UvrABCD factors are required for the NER pathway[2]. UvrAB form a complex at the sites of DNA damage that recruits UvrC endonuclease to cut the damaged strand on each side of the lesion[3,4]. DNA polymerase I (with or without UvrD helicase) then displaces the excised 12–13 nt long oligomer and fills in the gap using the complimentary ssDNA strand as a template[5].

Lesions repaired by NER produce a wide diversity of DNA structures[6] that must be promptly distinguished from the bulk of undamaged DNA. Currently, most NER is thought to occur through Global genome repair (GGR), a pathway where UvrAB discovers a lesion via random encounters with the bacterial chromosome[7]. Transcription-Coupled Repair (TCR) has been defined as a sub-pathway of NER where damage recognition is facilitated by the elongating RNA polymerase (RNAP), which stalls at bulky DNA adducts[8–10]. TCR is thought to occur only at the template strand (TS) of transcriptionally active regions based on the original observation that the TS is repaired faster than the non-template strand (NTS) of a gene[11].

TCR requires the aid of a protein factor to move RNAP away from the lesion site, making the site accessible to the UvrAB complex. The Mfd protein was first postulated to be such a TCR factor based on in vitro observations[12]. Subsequent in vitro experiments led to the proposal of the Mfd-mediated TCR mechanism where Mfd pushes RNAP forward at a lesion site, terminates transcription, and recruits the UvrAB complex to the damage site[13–15]. It is widely believed that, in the absence of Mfd, only the slower GGR pathway of NER will occur. In addition, because Δmfd cells are almost as sensitive to genotoxic stress as wild type parent strains[16–18] and exhibit no obvious deficiency in repairing bulk of their DNA[19], TCR is viewed as a minor accessory pathway that is not crucial for the removal of most of the lesions.

Recent studies have shown that TCR can occur in an Mfd-independent manner, through the concerted action of UvrD and NusA in complex with RNAP to promote backtracking of an RNAP stalled at a lesion site[16,20] and that the molecular alarmone ppGpp facilitates this process[21,22]. The discovery of the alternative TCR pathway suggests that TCR may have a larger role during NER than initially thought.

As most TCR studies have been done using low throughput techniques or biochemical approaches, there are very few experiments that address the role of transcription in NER at a genome-wide level. A next generation sequencing (NGS) assay called XR-seq was recently used to track repair through direct sequencing of the 13-mer oligo excised by NER[23,24]. The authors of these studies concluded that Mfd is necessary and sufficient for TCR, thus reinforcing the notion that TCR is only a minor sub-pathway of NER in bacteria. However, the conclusion from these XR-seq studies had a major limitation in their analysis by only focusing on the repair ratio of the TS/NTS. This ratio can decrease not only from a decrease in TS repair, but also an increase in NTS repair. Furthermore, the transcription levels were not determined under the conditions of XR-seq experiments. Therefore, a proper analysis of the repair as a function of transcription of each strand has not been performed. In addition, tracking repair at several different recovery timepoints from the same sample is not possible with XR-seq and it is, therefore, repair kinetics cannot be determined.

Here, we adopted a high-throughput single nucleotide resolution NGS method to directly monitor the appearance and repair of UV-induced CPDs over time throughout the E. coli chromosome. In contrast to the current dogma, and in agreement with

our mechanistic studies (Bharati et al.)[41], we demonstrate that transcription is coupled to the majority, if not all, of NER events and that this pervasive TCR process is largely Mfd-independent.

## Results

**CPD-seq measures genome-wide repair in E. coli.** We used a modified version (Fig. 1a) of the CPD-seq assay originally developed to study NER in yeast[25] to map CPD lesions in E. coli genomic DNA prior to and after exposure to UV light. We find that samples collected immediately after UV exposure show a dose-dependent ($60 \, J/m^2$ and $240 \, J/m^2$) enrichment of sequencing reads that were adjacent to dipyrimidine sequences (Fig. 1b). Reads associated with TT sequences were the most abundant after UV in comparison to CT, TC, and CC sequences. Samples collected at recovery timepoints displayed a decrease in all four dipyrimine sequences, indicative that repair by NER has occurred over time (Fig. 1b). Because lesions occurring at TT sites provided the most enrichment above non-dipyrimidine sequences at both UV doses, we focused our analysis on reads associated with TT sequences.

**Active transcription is required for GGR.** As transcription is more pervasive in bacteria than initially thought[26–29], we hypothesized that NTS and intergenic regions may be subject to TCR. To determine the contribution that active transcription has on the repair of each strand in the genome, we performed CPD-seq in the absence and presence of the antibiotic rifampicin (Rif), which inhibits promoter escape by bacterial RNAP. Because RNAPs that are already ongoing transcription elongation are not inhibited by Rif, we allowed a 1 h exposure time with Rif to ensure that vast majority of elongation complexes (ECs) would no longer remain on the genome. We find that, in the presence of Rif, both the TS and NTS are severely compromised in repair compared to cells not exposed to Rif (Fig. 1c). The negative effect of Rif on NER was comparable to that of genetic inactivation of UvrA and UvrD (Fig. 1c).

The Rif conditions we used would prevent the induction of several core NER factors derepressed during the SOS response[30,31]. Therefore, we also performed CPD-seq in a lexA3 mutant strain, which expresses a cleavage resistant variant of the LexA repressor[32], to determine whether the lack of SOS induction was responsible for the extreme repair deficiency we observed in the presence of Rif. To correlate this data with transcription, we performed strand specific RNA-seq in cells exposed to UV. After splitting genes based on their sense transcription level (high, mid, and low, see Methods), we find that recovery in the lexA3 mutant cells occurs to a similar extent as WT cells (Fig. 1d), except for a moderate deficiency in repair in the NTS of all genes and the TS of low transcribed genes (Fig. 1d). To be sure that NER factors were not degraded during the temporary Rif exposure, we performed Western Blots with and without a 2 h Rif exposure and find that Rif does not diminish the cellular protein levels of UvrA,B,C or D during the course of the experiment (Supplementary Fig. 2). Together, these results demonstrate that active transcription on both the TS and NTS is required for repair of most UV-induced lesions.

**NTS and intergenic regions are subject to TCR.** To further explore the relationship between transcription and NER, we used our RNA-seq data described above to directly correlate transcription data with repair data from CPD-seq under the same experimental conditions. This analysis revealed that the sense transcription level of a gene determines the amount of TS repair preference that occurs over the NTS. High transcribed genes have

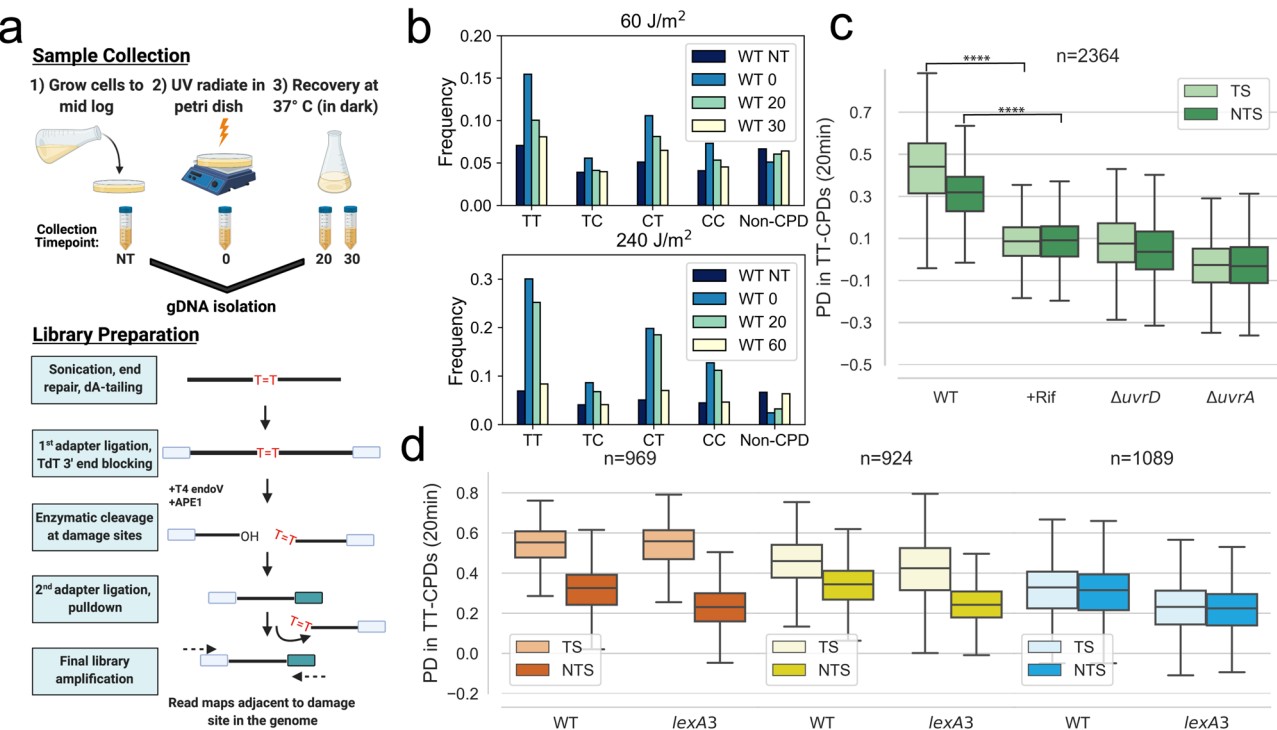

**Fig. 1 Active transcription is required for the recovery of TS and NTS from CPD-lesions. a** Schematic showing sample preparation and construction of CPD-seq libraries. **b** The frequency of reads that were adjacent to a potential CPD lesion site (TT, TC, CT, and CC) or a Non-CPD site for each recovery timepoint. Data is shown for a low (60 J/m²) and high (240 J/m²) doses of UV. **c** The recovery of lesions after 20 min in WT, Rif-treated WT, ΔuvrA and ΔuvrD cells (****$p < 10^{-308}$). Boxplots show the distribution of the percent decrease in TT-CPDs for each strand in gene bodies that had at least 1.5-fold TT-CPD enrichment at the 0-timepoint over NT-timepoint (see methods, Supplementary Fig. 1). **d** The recovery of lesions after 20 min in WT and lexA3 mutant cells. Genes were split by transcription level: high = orange, mid = yellow, and low = blue (for details on how genes were split, see Methods). The box shows the interquartile range (IQR), the line shows the median, and the whiskers extend to 1.5xIQR. The top of the box represents the 75th percentile and bottom represents the 25th percentile. *n* denotes the number of genes in each plot. *P*-values were calculated using a paired Wilcoxon signed rank test.

a high TS repair preference (Fig. 2a; left panel) at early repair timepoints (10 and 20 min) and this preference decreases in mid transcribed genes (Fig. 2a, middle panel). Moreover, low transcribed genes do not display a repair preference for either strand (Fig. 2a; right panel). This analysis clearly shows that TCR occurs genome-wide, but only highlights TCR that occurs in sense regions because we categorized genes by their sense transcription.

The NTS is repaired at the same rate in all three gene categories (Fig. 2a), however, the rate of repair drastically decreases in these regions after exposure to Rif, (Fig. 1c) meaning that TCR does occur in "NTS" regions as well. To determine if antisense transcription correlates with NTS repair, the TS/NTS repair ratio was determined for every gene using CPD-seq data. Genes were categorized in high, mid, or low repair groups based on the TS/NTS CPD-seq ratio, meaning genes with high TS repair preference were considered high repair whereas genes with a NTS repair preference were considered low repair. The distribution of the sense/antisense transcription ratio (TS/NTS RPKM) was then determined for every repair category and plotted (Fig. 2b). From this analysis, we find that repair and transcription are correlated on a genome-wide level. We also observed that many genes with a low TS/NTS repair ratio (or a NTS repair preference) also showed a low sense/antisense ratio (Fig. 2b). We then asked if genes with greater antisense over sense transcription would show a repair preference for their NTS. Indeed, when we calculate the percent decrease in CPDs in 122 genes that showed high antisense over sense transcription, we find a repair preference in the NTS over the TS (Fig. 2c). Thus, it appears that the local level of transcription determines the rate of NER in the NTS, the same way as it does in the TS.

However, this analysis used the gene body to determine antisense transcription preference and a limited number of genes met these criteria. We predicted that most antisense transcription would occur at the 3' end of a gene due to readthrough from a neighboring gene in the opposite direction. Indeed, the further RNAP advances while transcribing the antisense strand, the higher the probability it would be terminated by Rho[27,29,33]. We, therefore, determined which genes have an antisense transcription preference in the last 100 bp and 50 bp downstream into intergenic regions. We find that twice as many genes have at least a 2-fold antisense transcription preference at their gene end as opposed to the entire gene (276 vs. 122). A meta-analysis of the CPD lesions in these genes at the 0 min timepoint and 10 min timepoint revealed that the NTS recovered faster at the end of these gene bodies versus the beginning, which confirms that the higher RNAP presence on the NTS at gene ends led to increased repair in these regions (Fig. 2d). This analysis was performed on varying gene end windows and produced a similar result as the 100 bp upstream and 50 bp downstream gene end window (Supplementary Fig. 3). The above analyses provide evidence that TCR occurs on antisense and intergenic regions and suggest that rates of repair are primarily determined by RNAP location on the genome.

**Global antitermination improves both TS and NTS repair.** If the presence and amount of transcription determines how much repair will occur in a given genomic region, we hypothesized that we could accelerate repair of NTS regions by increasing the level of genome-wide antisense transcription. Previous studies have

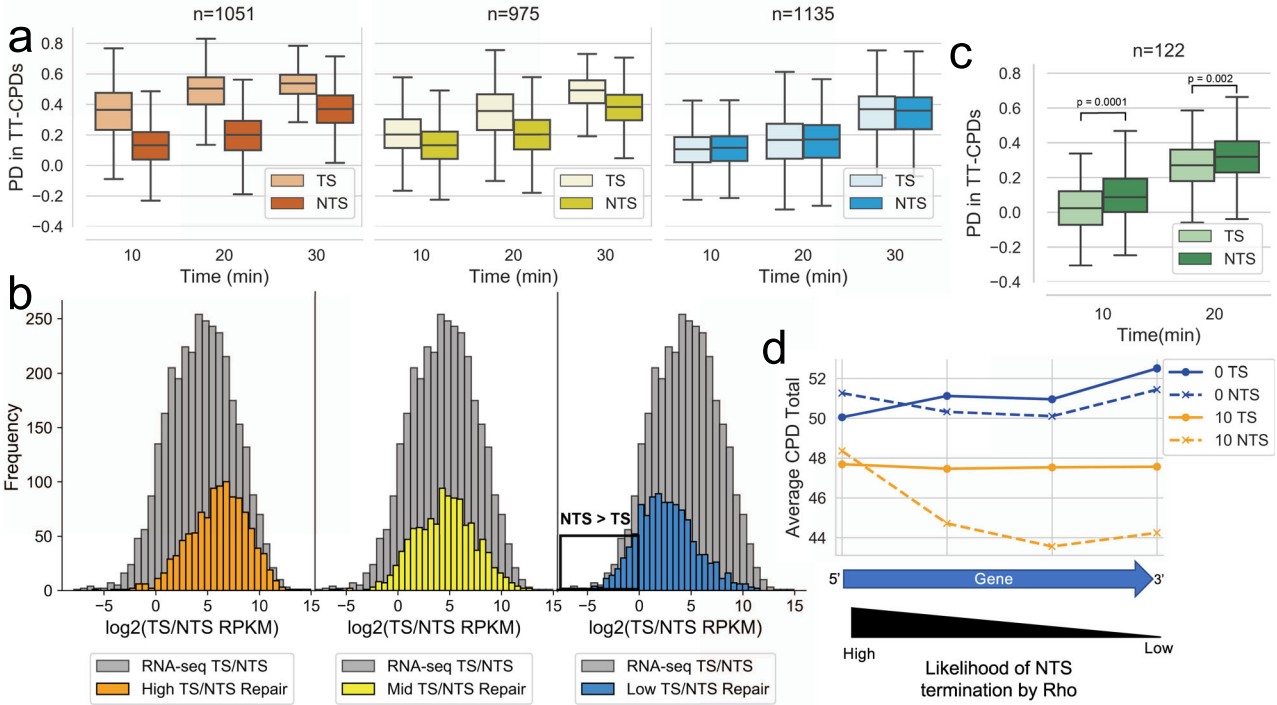

**Fig. 2 TCR occurs in sense, antisense, and intergenic regions. a** The recovery of TT-CPD lesions at 10, 20, and 30-min recovery timepoints in WT cells. Genes were split by transcription level based on RNA-seq data: high = orange, mid = yellow, and low = blue (for details on how genes were split, see Methods). **b** The sense/antisense (TS/NTS) transcription ratio of high (orange), mid (yellow) and low (blue) repair genes. Genes were split into three equal repair levels based on their TS/NTS percent decrease (PD) ratio from CPD-seq data after 20 min of recovery. The grey histogram represents the TS/NTS transcription ratio of all genes. The black box highlights genes with an antisense transcription preference. **c** The recovery of TT-CPD lesions in genes that have at least a 1.5-fold antisense preference. The box shows the interquartile range (IQR), the line shows the median, and the whiskers extend to 1.5xIQR. The top of the box represents the 75th percentile and bottom represents the 25th percentile. *n* denotes the number of genes in each plot. *P*-values were calculated using a paired Wilcoxon signed rank test. **d** Meta-analysis displaying the TT-CPD total in genes with an antisense transcription preference at their gene end at the 0 min and 10 min timepoint. Each point represents a window encompassing a quarter of the gene.

shown that termination factor Rho suppresses pervasive transcription[26,27,29,33]. To test if increased readthrough leads to increased repair, we performed CPD-seq in two Rho mutant strains, each with one single amino acid substitution in Rho (L187R and D210G). These mutants have been shown to cause readthrough past a Rho termination site in a reporter gene[34]. RNA-seq of both D210G and L187R show that readthrough occurs genome wide and at many different Rho termination sites in both strains (Fig. 3a). Notably, each mutant has a different level of increased readthrough, with L187R showing stronger readthrough and a greater increase in NTS transcription levels than D210G (Fig. 3a, b). However, the relatively mild readthrough presented in these two strains (Fig. 3a) was enough to change the genome wide repair profile by increasing the rate of repair in NTS regions (Fig. 3c). Moreover, the increased repair is directly proportional to the amount of readthrough that occurs, with L187R having greater improvement in repair of NTS regions compared to D210G (Fig. 3c).

From this analysis, we also observed that both Rho mutants were increasing repair in TS regions. This increase in TS repair was due to genes that showed an increase in sense transcription as a result of the mild Rho inhibition (Supplementary Fig. 4). For an independent confirmation that these NTS regions are commonly affected after increased readthrough at Rho terminators, we followed 1563 NTS regions that increased in L187R compared to WT (Supplementary Fig. 5a) in a separate Rho mutant known to increase readthrough (*rho15*)[35]. We found that *rho15* cells also showed improved repair of the NTS of these regions (Supplementary Fig. 5b). These results confirm that

global transcription readthrough leads to a corresponding increase in repair.

**UV-induced genotoxic stress promotes global transcription antitermination.** The classical model of NER/TCR proposes that the NTS would mostly be repaired through GGR, a mechanism that does not involve elongating RNAP for damage detection. However, because we observe that Rif almost completely abolished NER in NTS regions, and that even mild increases in transcription readthrough can have a global repair benefit in NTS regions, the damage in a majority of NTS and intergenic regions must be detected through TCR. We thus hypothesized that the genotoxic stress may prompt the increased RNAP presence in these low transcribed regions. To test this conjecture, we performed RNA-seq in *E. coli* cells exposed to UV and determined whether there was an increase in transcription of NTS regions similar to what we observed in the Rho mutants described above. Remarkably, we observed a global increase in antisense transcription in WT cells after UV exposure (Fig. 4b). After performing genome-wide readthrough analysis at Rho-dependent terminators, we find that UV promoted global Rho antitermination (Fig. 4b, Supplementary Fig. 7). UV-induced antitermination at selected Rho-dependent terminators[36] (Supplementary Fig. 6) was confirmed by RT-qPCR and found to be proportional to UV dosage (Fig. 4c, d). A likely explanation for why readthrough might increase in response to UV is that the SOS response causes changes that inhibit termination by Rho. This hypothesis explains why the deficiency in SOS response

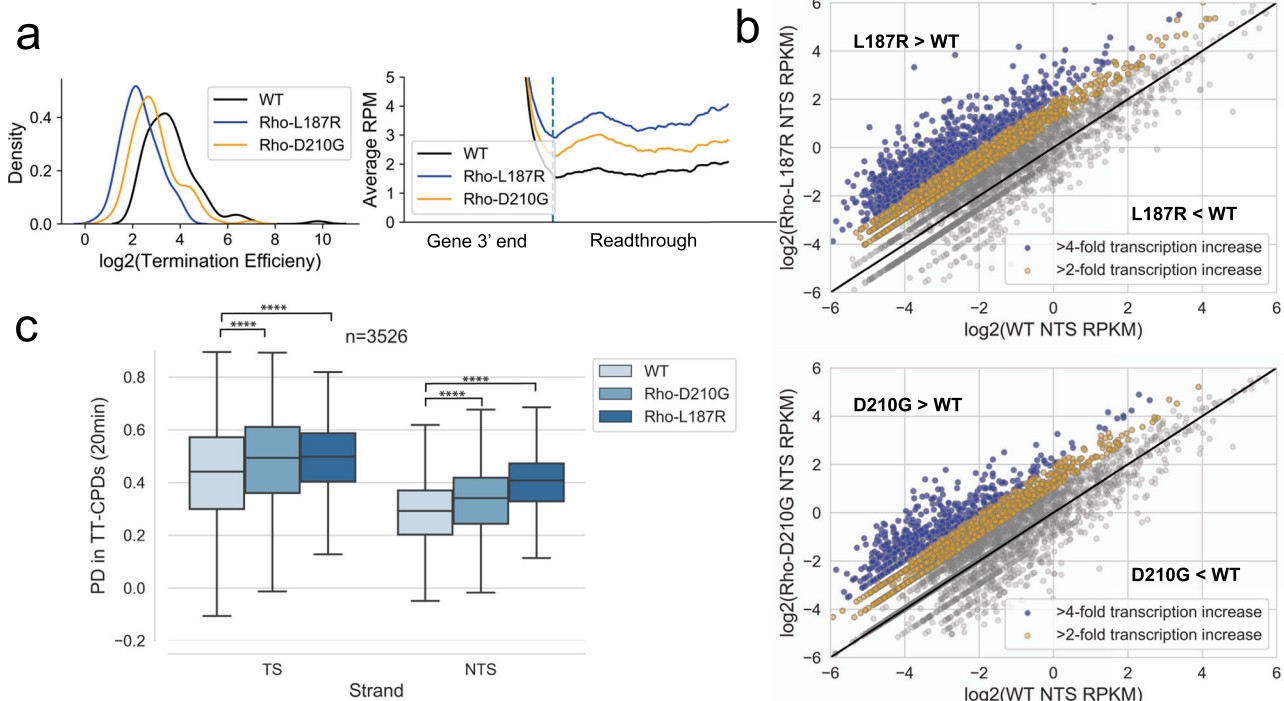

**Fig. 3 Global antitermination improves genome-wide NER. a** The distribution of the $\log_2$ termination efficiency (see Methods) for each Rho-dependent terminator in WT, Rho$^{D210G}$ and Rho$^{L187R}$ strains (left panel). Meta-analysis of the average readthrough past Rho-dependent termination sites in WT, Rho$^{D210G}$ and Rho$^{L187R}$ strains. The dashed line represents the termination site (right panel). **b** Scatterplots comparing the NTS RPKM (RNA-seq) in WT vs Rho$^{L187R}$ (top) or Rho$^{D210G}$ (bottom) mutants. Genes are colored based on their fold increase in the Rho mutant compared to WT. Blue genes had a greater than 4-fold increase in antisense transcription and yellow genes had a 2-4-fold increase in antisense transcription. **c** The recovery of TT-CPD lesions in WT, Rho$^{D210G}$, and Rho$^{L187R}$ strains (****$p < 10^{-77}$). The box shows the interquartile range (IQR), the line shows the median, and the whiskers extend to 1.5xIQR. The top of the box represents the 75th percentile and bottom represents the 25th percentile. $n$ denotes the number of genes in each plot. $P$-values were calculated using a paired Wilcoxon signed rank test.

(*lexA3*) compromised repair primarily in the NTS, but not in the TS (Fig. 1d): Rho antitermination would mainly benefit the poorly transcribed antisense (NTS) and intergenic genomic parts by boosting TCR in those regions.

The architecture of the TCR complex (TCRC) may also explain the molecular mechanism of SOS-mediated Rho antitermination. Our recent findings establish RNAP as the platform for the assembly of the NER complexes containing UvrABD (Bharati et al.)[41]. The interacting surface between NER factors and RNAP overlaps with that of Rho and its cofactor NusG (Bharati et al.)[41]. As direct and persistent binding of Rho and NusG to RNAP is critical for the mechanism of termination[37–39], the TCRC is expected to be more resistant to termination. In support of this idea, we find that the overexpression of UvrAB leads to a similar increase in readthrough as UV at a representative Rho-dependent terminator (Fig. 4d, e).

**The role of Mfd in TCR.** Our data shown so far supports a model where transcription is an essential process required for repair of the entire genome. Mfd was recently proposed to be necessary and sufficient for all TCR that occurs in *E. coli*[23,24]. However, Mfd null cells display minimal sensitivity to UV and DNA damaging agents[16,18,21,40] and overall rapid recovery from UV-induced damage occurs in the absence of Mfd[19](Bharati et al.)[41]. Given our model of TCR (see also Bharati et al.)[41], we performed CPD-seq in an Δ*mfd* strain with and without Rif to determine the role Mfd plays in total recovery of the genome. We find that deletion of *mfd* in the absence of Rif slows repair of the TS compared to WT and that this deficiency is observed in mostly high transcribed genes (Fig. 5a and Supplementary Fig. 8). After

addition of Rif to an Δ*mfd* strain, we observed a major decrease in repair efficiency in both strands compared to Δ*mfd* without Rif (Fig. 5a). Because we find that TCR is globally important for repair of both the TS and NTS, this result implies that Mfd is not important for TCR that occurs in a majority of the genome.

Mfd is not an abundant protein, with an estimated number of Mfd molecules per cell being ten times smaller than the number of RNAP molecules engaged in elongation[41–43]. In addition, Mfd is constitutively expressed and not induced upon UV stress[30]. Thus, it must be very limited in the number of ECs before and after stress. The traditional model of TCR postulates that Mfd recruits UvrA to the lesion sites following the termination of stalled ECs[14,15]. If the TS repair deficiency that we observe in Δ*mfd* cells were due to a lack of Mfd-mediated UvrA recruitment, one can expect that providing the cell with additional Mfd would enhance the repair of these TS regions. However, the over-expression of Mfd globally inhibits repair of both the TS and the NTS of many genes (Fig. 5b), signifying that Mfd cannot be directly involved in recruitment of NER factors to DNA.

The deficiency in repair of highly transcribed regions that we observe in Δ*mfd* cells may be due to the role of Mfd in terminating regular elongation complexes (ECs) ahead of the dedicated TCR complexes (TCRC), which would otherwise obstruct TCRC access to the lesion sites (Bharati et al.; Supplementary Fig. 9a). However, similar to Rho, Mfd may compromise repair at low transcribed regions. Congruently, we performed RNA-seq in Δ*mfd* cells exposed to UV and observe a further increase in transcription readthrough compared to UV exposure alone (Fig. 5c), as well as an increase in global antisense transcription (Fig. 5d). A subset of 272 genes that increase in NTS

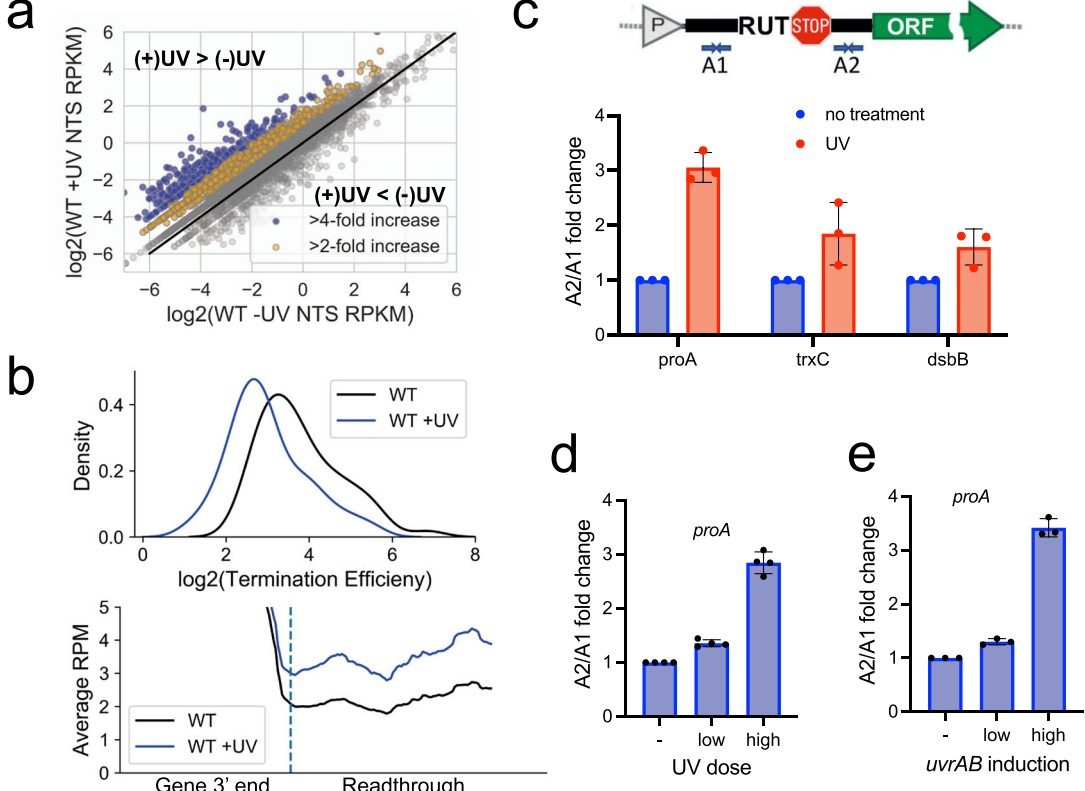

**Fig. 4 UV-induced genotoxic stress increases global antitermination. a** Scatterplot comparing the NTS RPKM (RNA-seq) in WT vs WT + UV (120 J/m²) cells. Genes are colored based on their fold increase after UV exposure compared to without UV. Blue genes had a greater than 4-fold increase in antisense transcription and yellow genes had a 2-4-fold increase in antisense transcription. **b** The distribution of the $\log_2$ termination efficiency (see methods) for each Rho-dependent terminator in WT and WT + UV cells (top). Meta-analysis of the average readthrough past Rho-dependent termination sites in WT and WT + UV cells. The dashed line represents the termination site (bottom). **c** Schematic showing the location of the qPCR primers A1 and A2 relative to the RUT and termination site (top). The A1/A2 fold-change at three different native Rho terminators after UV exposure (bottom). **d** The A1/A2 fold change at the proA terminator after exposure to a high or low UV dose. **e** The A1/A2 fold change at the proA Rho-depended terminator after high or low UvrAB induction. Values are means ± SD from three independent replicates.

transcription also show *increased* NTS repair in an Δ*mfd* mutant and these regions also show improved repair in the two Rho mutants described in Fig. 3 (Fig. 5e).

## Discussion

The classical model of NER states that GGR and TCR are two distinct damage recognition mechanisms and that TCR is not essential to the recovery from DNA damage. GGR has been viewed as the predominant means by which NER would act. This is counterintuitive though, as UvrA lacks its own motor function to scan DNA in one dimension and has to rely on 3D search to detect rare lesions in bulk of undamaged DNA[2]. In vivo, this process must be dramatically complicated by DNA compaction, molecular crowding, numerous DNA binding proteins, and by the shear amount of competitive intact DNA for which UvrA has almost the same affinity as to nondamaged DNA[44,45]. The only reason why TCR was deemed a minor sub-pathway of NER is because of Mfd, which is considered to be the necessary and sufficient TCR factor in bacteria[24], yet with a marginal phenotype[16–18].

Here, we provide evidence that transcription elongation is required for repair of lesions caused by UV genome-wide, at least in *E. coli*. When we halted the majority of RNAP molecules at promoters by temporary exposing cells to high Rif, we see little to no recovery from UV-induced damage in both TS and NTS regions. Additionally, the combination of RNA-seq and CPD-seq allowed us

to observe that TCR occurs in antisense and intergenic regions. In fact, even mildly increasing RNAP presence in those genomic regions was enough to substantially accelerate their repair, proving that NER is highly sensitive to RNAP location on the genome. Overall, we observe that preferential repair of the TS strand does not occur because of a distinction between GGR and TCR, but is due to higher presence RNAP on the TS, resulting in faster repair than the NTS, which contains a smaller number of TCRCs. We argue that TCR occurs globally and is the major way that lesions are repaired through NER.

RNAP is likely to be present on the entirety of the genome given the findings that pervasive transcription is prevalent in bacteria[46]. The sites of pervasive transcription are not highly conserved between bacterial species[47], which has led to the conclusion that pervasive transcription is an unavoidable byproduct of gene expression rather than a phenomenon that has a systemic functional purpose. Our findings counter this idea by showing that bacteria can boost pervasive transcription in response to genotoxic stress via mild Rho inhibition to produce an increase of RNAPs in transcriptional "dead zones", allowing these regions to be scanned for DNA damage (Supplementary Fig. 9b). This increased transcriptional readthrough must be finely balanced, as too much of it can be toxic to bacteria[26,48], whereas too little would abolish repair. NER detects a wide variety of lesions in addition to CPD-lesions focused on here. However, many of these lesions such as, for example, 4NQO-derived purine adducts or N2-fufuryl-dG, are not as efficiently detected by the UvrAB complex[49] and, therefore, it is

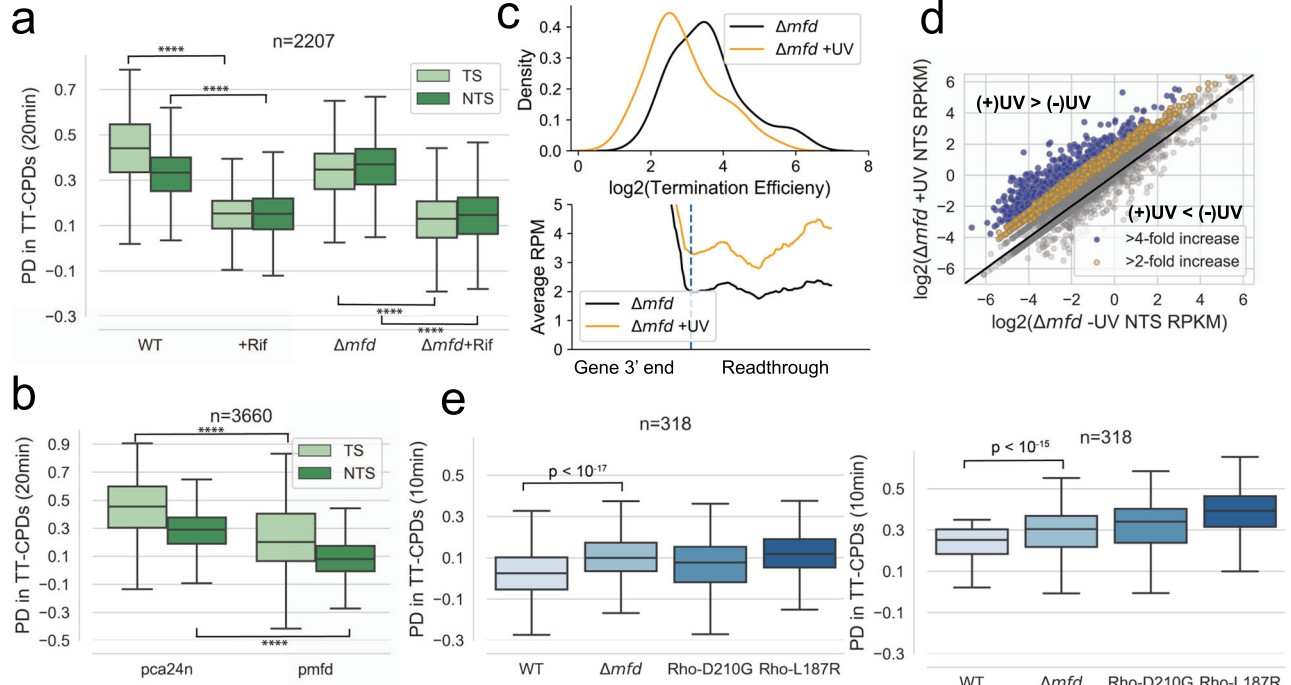

**Fig. 5 The role of Mfd in repair. a** The percent decrease in TT-CPDs of the TS and NTS of gene bodies in WT, WT + Rif, Δ*mfd* and Δ*mfd* + Rif. (****$p < 10^{-277}$). **b** The percent decrease in TT-CPDs of the TS and NTS of gene bodies in cells overexpressing Mfd. (****$p < 10^{-308}$). **c** The distribution of the $\log_2$ termination efficiency (see methods) at Rho-dependent terminators in the absence or presence of UV exposure in Δ*mfd* cells (top). Meta-analysis of readthrough (see methods) past Rho-dependent termination sites in the absence or presence of UV exposure in Δ*mfd* cells (bottom). **d** Scatterplot comparing the NTS RPKM (RNA-seq) in Δ*mfd* vs Δ*mfd* + UV (120 J/m$^2$) cells. Genes are colored based on their fold increase after UV exposure compared to without UV. Blue genes had a greater than 4-fold increase in antisense transcription and yellow genes had a 2-4-fold increase in antisense transcription. **e** The percent decrease in TT-CPDs of the NTS in gene bodies where Mfd deletion increases NTS transcription. The box shows the interquartile range (IQR), the line shows the median, and the whiskers extend to 1.5xIQR. The top of the box represents the 75th percentile and bottom represents the 25th percentile. *n* denotes the number of genes in each plot. *P*-values were calculated using a paired Wilcoxon signed rank test.

even less likely that GGR would have any role in their repair. RNAP presence through pervasive transcription may also aid in the detection of a wide variety of bulky lesions that must be removed from the genome (Bharati et al.)[41].

The precise molecular mechanism for UV-induced anti-termination remains to be elucidated. We predict that Rho activity may be compromised in low transcribed regions due to an increase in TCRCs that form during the SOS response. This hypothesis explains why SOS deficiency (*lexA3*) affects NER primarily in low transcribed regions of the genome, which are intrinsically deprived of TCRC. Our in vivo structural proteomics and biochemical data (Bharati et al.)[41] demonstrate that UvrA and UvrD bind directly to RNAP, even before damage exposure. Positioning of NER proteins in these TCRCs may interfere with NusG and/or Rho binding to RNAP, which is a critical step in Rho-dependent termination[37–39]. This idea is consistent by our observation that UvrAB overexpression increases antitermination at a representative Rho-dependent terminator (Fig. 4e). Exploration of this antitermination mechanism is a subject for future studies.

Given that TCR is prevalent globally, we re-examined the role that Mfd would play in this process. Our results align with previous studies showing that Mfd contributes to the rapid recovery of the TS of highly transcribed genes[23]. However, we found that Mfd does not contribute to repair of the NTS and of both strands in low transcribed genes and can even be detrimental to NER in many parts of the genome. If Mfd were a TCR factor, let alone the only factor required for TCR[24], we would expect that its absence would compromise NER globally, similar to Rif, and particularly in low transcribed genomic regions, which still depend on TCR.

Not only did Mfd fail to support TCR of low transcribed genes, its overexpression inhibited global NER. We propose that Mfd can be indirectly beneficial at highly transcribed regions because it helps clearing the path to DNA lesions by TCRC. Genes with high transcription are more likely to have an array of "regular" ECs stuck between the lesions and trailing TCRCs (Supplementary Fig. 9a, b). Mfd is capable of terminating multiple ECs in a row[50]. The directionality of Mfd is likely to be determined by UvrA, which is a part of the TCRC, that helps recruiting Mfd to DNA in vivo (Bharati et al.[41]. This model is consistent with most available genetic and biochemical data and establishes Mfd as a DNA "cleaning" factor that is recruited by UvrA, as opposed to a TCR factor that recruits UvrA in vivo. Akin to Rho, however, the activity of Mfd must be balanced not to terminate TCRCs as well, hence its ambiguous role in TCR and the minimal NER phenotype.

Although the work described here was performed in *E. coli*, NER is a highly conserved process and, therefore, we expect our model to be widely applicable to other bacteria who share the same key protein factors involved in NER. Additionally, pervasive transcription is well established and widespread in eukaryotes[51–53] indicating that RNAP II presence in antisense regions may play a crucial role in recruiting mammalian NER factors to lesion sites not only on highly transcribed TS regions but also in antisense and low transcribed areas. Genome-wide studies of NER in human cells claim that TCR can only occur on the TS through the Mfd analog, CSB (human)/rad26 (yeast)[54,55]. However, in these studies, the ratio of TS/NTS repair was used to assess whether TCR occurs, rather than using an analysis that assesses the role of transcription in repair in regions outside of the

TS. It is therefore likely that eukaryotes use transcription for efficient damage detection and repair globally, similar to pervasive TCR we describe here in *E. coli*.

## Methods

**Strains**. Strains used in this study can be found in Supplementary Table 1.

**Oligos**. A list of oligos used in this study can be found in Supplementary Table 2.

**UV conditions**. Overnight *E. coli* culture was grown at 30 °C in fully supplemented MOPS (Teknova). The next day, 600 μl of overnight culture was added to 60 mL of fully supplemented MOPS and was grown at 37 °C until it reached an $OD_{600}$ of ~0.4. In the case where rifampicin (Rif) treatment was used, 750 μg/mL of Rif was added for 1 h. The Rif treated cells were then spun down and resuspended in fresh fully supplemented MOPS media. 10 mL of culture was added to 10 mL of 2X NET buffer (200 mM NaCl, 20 mM EDTA pH 8, 20 mM Tris-Cl;pH 8) and kept on ice during the subsequent UV exposure step (NT timepoint). The rest of the culture (~50 mL) was transferred to a 140 × 20 mm petri dish and placed on a rotator so that the cells would continuously move around the petri dish. A UV (254 nm) lamp was used to expose the cells to the appropriate UV dose. Immediately after UV exposure, 10 mL of culture was added to 10 mL of NET buffer and placed on ice (0-timepoint). The remaining culture was transferred to a foil-wrapped flask and moved to a shaker at 37 °C for recovery. The cells for remaining timepoints were centrifuged and pelleted. All cell pellets were stored at -80 until genomic DNA extraction.

**Rifampicin treatment and quantitation of RNAP and NER players using semi-quantitative Western Blot**. To quantitate the proteins involved in NER before and after rifampicin exposure, UvrA, UvrB, UvrC, and UvrD were chromosomally tagged with 3X-FLAG at their C-terminal in the parent strain of *E. coli* MG1655, whereas the beta-prime subunit of RNAP (*rpoC*) was 6X-His tagged[41]. All the derivative strains were grown in LB-broth at 37 °C overnight and sub-cultured into a fresh LB-Broth media until an $OD_{600}$ of 0.3 ± 0.05. Rifampicin (750 μg/mL) was added to the cultures and incubated for an additional 90 min. The cells were centrifuged and washed twice with cold 1X PBS and stored at -80 freezer for further use or lysed in lysis buffer (500 mM NaCl, 50 mM HEPES; pH 7.5, and 5% glycerol) supplemented with protease cocktail inhibitor using lysozyme (2 mg/mL) treatment and sonication. The lysed samples were centrifuged at 20,800 x g (14,000 r.p.m.) for 10 min at 4 °C to remove the cell debris. For the semi-quantitative Western blots, 25 μg of cell lysates were resolved on SDS-PAGE and transferred to PVDF membrane. The membranes were blocked with PBS-T (PBS;pH 7.4 with 0.05% Tween-20) containing 5% skim milk for 1 h at room temperature, followed by incubation with commercially available monoclonal anti-FLAG M2 (Sigma-Aldrich; F1804) and anti-6X His tag antibodies (Abcam; Ab9108). The fluorescent probe conjugated secondary antibodies, Alexa Fluor 488 donkey anti-mouse IgG (Invitrogen, A-21202) and Donkey anti-Rabbit (Alexa Fluor 647, Invitrogen, A-31573) were used and developed with FluorChem R. The intensities of the corresponding protein bands for RNAP (β'-subunit) and 3X-FLAG tagged UvrA, UvrB, UvrC, and UvrD were measured using the ImageJ 1.51k[56]. To normalize the intensity, the ratio of rifampicin treated samples were divided by the ratio of the control (mock treated) sample and represented as % intensities.

**CPD-seq library preparation**. Genomic DNA was purified from *E. coli* using Human Lysozyme (Sigma) and the Monarch Genomic DNA Extraction kit (NEB). 5 μg of total genomic DNA was sonicated using the Covaris for 1 min and 5 sec (Duty Factor: 20%, Peak Incident Power: 50 W, Cycles Per Burst: 200) to create DNA fragments averaging 400 bp. The fragmented DNA was then purified with 1.2X PCRClean beads (Aline). The fragmented DNA was end prepped by first treating with NEBNext End Repair Module, cleaning with 1.2X beads and second, treating with NEBNext dA-tailing Module and cleaning with 1.2X beads. Adapter A1_T and A1_B were annealed by incubating an equal molar amount at 95 °C for 5 min and gradually cooling to room temperature. The adapter was then ligated to the end prepped DNA sample O/N at 16 °C using the NEBNext Quick ligation module and cleaned the next day with 1.8X beads. Adapter ligated DNA was treated with Terminal Transferase (NEB) to block any remaining free 3' ends and cleaned with 1.8X beads. The sample was then treated with 10 units of T4 PDG (NEB) to cleave any CPDs in the DNA and cleaned with 1.8X beads. Treatment with 10 units of APE1 (NEB) and 5 units of rSAP (NEB) was then performed to create a 3'OH group at the CPD cleavage site. A2_T and A2_B were annealed for 95 °C for 5 min with gradual cooling. Before the next adapter ligation step, the sample was heated at 95 °C for 5 min to separated DNA strands. After, the second adapter was ligated using the NEBNext Quick Ligation Module O/N at 16 °C. Any samples that contained the second adapter were then pulled down with Streptavidin M-280 Dynabeads (Thermo Fisher) and the ssDNA was released with 0.15 M NaOH. The final library was then amplified using NEBNext High Fidelity 2X PCR Master Mix. Libraries were quantitated on a Qubit and QC was performed on an Agilent 4200 Tapestation. Libraries were sequenced on an Illumina NextSeq using a high output 75 cycles kit (paired end).

**CPD-seq analysis**. Bowtie[57] was used to align read 2 to the *E. coli* (U00096.3) reference genome using the parameters -S -m1 –seed 123. Reads were then split by orientation using Samtools[58]. The position of the last nucleotide in each read was determined using custom Python Scripts. Reads that corresponded to a TT-site in the genome were used for further analysis. The Percent Decrease (PD) in CPDs over time was calculated by determining the number of TT-reads (normalized by sequencing depth and TT-sites per gene) mapped to each gene at the 0-timepoint as well as any subsequent timepoints (10, 20, 30 min). Genes that did not have a 1.5-fold increase in counts at the 0-timepoint over the NT timepoint were removed from the analysis. Because these genes did not have increased signal after damage exposure, tracking their recovery would add noise to the analysis. The majority of genes passed this filter threshold (Supplementary Fig. 1). The TT-reads at a subsequent timepoint was then subtracted from the TT-reads at the 0-timepoint and this value was divided by the TT-reads at the 0-timepoint in order to obtain the final PD value.

**Transcription level boxplots**. The results from the percent decrease analysis were split into three different categories based on the RPKM of the gene as calculated from the WT + UV RNA-seq data. Genes were split into the following categories: high transcription – RPKM > 30, mid transcription - RPKM > 5 and RPKM < = 30, and low transcription – RPKM < = 5.

**RNA-seq library preparation**. Overnight culture was diluted 1:100 in 10 mL of LB media. Cells were grown to $OD_{600}$ ~0.4. 1 mL of cells was collected in 1.5 mL tube for the -UV sample. The remaining cells were transferred to a small petri dish and exposed to UV (120 J/m²) on a rotator that allowed continuous shaking. 1 mL of UV exposed cells was transferred to a 1.5 mL tube and both the -UV and +UV samples were allowed to recover in the dark for 5 min at 37 °C. The cells were then immediately pelleted and stored at -80 °C for RNA extraction. RNA was then extracted from cell pellets using the Masterpure Complete DNA and RNA Purification Kit (Lucigen). RNA was then treated with TURBO DNA-Free Kit to remove any remaining genomic DNA from the sample. Ribosomal RNA was depleted from the samples using the Ribominus Transcriptome Isolation Kit (Thermo Fisher). RNA-Sequencing libraries were prepared using the rRNA depleted RNA and the NEBNext Ultra II Directional RNA Library Prep Kit (NEB). Libraries were sequenced on an Illumina NextSeq using a high output 75 cycles kit (paired end).

**RNA-seq analysis**. Reads were aligned using Bowtie2[59] with the parameters –quiet, -p 2, -x, and using the U00096.3 reference genome (NCBI). The aligned reads were then split based on whether they were in the forward or reverse orientation with Samtools[58]. HTseq[60] was used to determine the amount of reads that mapped to each gene and Python scripts were used to generate the final count tables of reads per strand for each gene. The Reads Per Kilobase Million (RPKM) was then calculated for each sample using these counts. Readthrough analysis was done using bedtools[61] genomecov to determine the read coverage across the entire *E. coli* genome. These counts were then normalized by the total number of aligned reads per million (RPM). Meta-analysis: The RPM for a given window around specified Rho termination sites was then determined and averaged across all sites in the analysis. Termination efficiency: The positions of Rho-dependent termination sites were obtained from a previous study[36]. For every Rho dependent termination site, the sum of reads in the windows before and after the termination site was calculated. The ratio was then calculated using the sums from these two windows and was considered the termination efficiency for that Rho-dependent terminator.

**mRNA purification, reverse transcription and qPCR**. MG1655 strain was inoculated from fresh LB plates into 3 mL of LB media and was grown overnight at 37 °C with shaking. Next day (~20 h) 30 mL of fresh LB were inoculated by 0.3 mL of overnight culture and grown in 250 mL flask at 37 °C with shaking till OD600~0.3. Two 10 mL aliquots were taken from the culture and one was transferred into a standard Petri dish and another one was not. Sample in the Petri dish was irradiated at room temperature with UV 192 J/m2 with shaking. Both irradiated and unirradiated samples were transferred into 15 mL Falcon tubes before spinning 5 min at 5000 g at 4 °C. The supernatant was discarded and mRNA was purified using MasterPure Complete DNA and RNA purification kit (Lucigen) according to the manufacturer manual except that DNase I treatment was conducted 30 min at 37 °C and was supplemented with 500 units of ExoIII and 25 units Sau3AI (both from New England Biolabs). RNA was re-dissolved in 50 μL TE buffer (1–2 mg/mL) and then volume was adjusted to 500 ng/μL by TE. cDNA was produced from 1 μg RNA using Qiagen QuantiTect reverse transcription kit according to the manufacturer. For strand specificity, appropriate 0.7 μM reverse primers were used instead of the random primers provided by the kit. qPCR was performed using QuantStudio 7 Flex real time qPCR machine (Applied biosystems) from 5 ng DNA in each well (20 μL per well) in triplicates using appropriate gene specific primer pairs (1 μM each). Annealing temperature was 60 °C. qPCR primers used in this work can be found in Supplementary Table 3.

**Rho termination experiment**. For Rho termination assay transcription was performed using DNA templates consisting of T7A1 promoter fused to appropriate

Rho terminator sequence (see below). DNA templates were produced using PCR reaction and purified from agarose gel with Quiagen gel-extraction kit according to the manufacturer. To initiate the reaction, 7.5 pmol core RNAP was mixed in 20 μL TB100 buffer (40 mM Tris-HCl, pH 8.0; 10 mM MgCl$_2$; 100 mM NaCl) with σ$^{70}$ (10 pmol) subunit, and 7.5 pmol appropriate DNA template. The reaction was incubated at 37 °C for 5 min. 10 μM ApUpC RNA primer plus 25 μM GTP and ATP were added to the sample and incubation continued at 37 °C for 5 min. Next, reactions were mixed with 2 μl α-[$^{32}$P]-CTP (Perkin Elmer) and incubated at room temperature for 5 min, followed by addition of non-radioactive 5 μM CTP. Incubation continued for 2 more min at room temperature. The samples were mixed with 20 μL TB100 and split into three 10 μL aliquots. Samples were mixed where indicated with 100 nM Rho hexamer and 1 μM NusG and incubated at room temperature for 5 min. Reactions were chased by 0.1 mM NTPs for 5 min at 37 °C before taking 10 μL aliquots and quenching them with 10 μL of stoppage buffer SB (1× TBE, 20 mM EDTA; 8 M urea, 0.025% xylene cyanol, and 0.025% bromophenol blue). Quenched samples were heated at 100 °C for 5 min and loaded onto 6% (20×20 cm) 19:1 polyacrylamide gel supplemented with 7 M urea and 1× TBE. The gel was run at 50 W for 20 min. The gel was transferred onto Whatman paper and dried at 80 °C for 30 min. The gel was exposed to a phosphor screen for 60 min and visualized by Typhoon Phosphorimager (GE Healthcare). DNA templates used in the transcription assays can be found in Supplementary Table 4.

**Isolation of BCM-resistant Rho mutants**. An overnight culture was split into 20 separate 1 mL cultures and grown overnight in LB containing 100 μg/mL bicyclomycin (BCM). Each overnight culture was then plated on LB or MOPs agar plates. Colonies were picked from any plates that showed growth and grown overnight in LB containing 100 μg/mL BCM. Colony PCR was performed on the overnight cultures that had growth. Isolation and sequencing of the PCR products was performed to confirm a mutation in the *rho* gene body.

**Reporting summary**. Further information on research design is available in the Nature Research Reporting Summary linked to this article.

## Data availability
The data supporting the findings of this study are available from the corresponding authors upon reasonable request. The DNA and RNA sequencing data generated in this study have been deposited to the Gene Expression Omnibus (GEO) under the accession code GSE17192. Source data for the figures and supplementary figures are provided as a Source Data file. Source data are provided with this paper.

## Code availability
Custom codes used for the analyses in this study have been uploaded and made publicly available through Zenodo (https://zenodo.org/record/5979538).

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

## Acknowledgements

We thank Aviram Rasouly for discussions on experimental design and data analysis. We thank Venu Karmarthapu for help with initial UV experiments and scientific discussions. We thank Ilya Shamovsky for processing libraries for Illumina sequencing and experimental design discussions. Schematic figures were made using BioRender.com. This work was supported by NIH grants F31 GM131516-02 (B.M.) and R01 GM126891 (E.N.), the Blavatnik Family Foundation, and by the Howard Hughes Medical Institute (E.N.).

## Author contributions

B.M. designed and performed CPD-seq, RNA-seq and Rho mutant experiments. B.M. performed data analysis. B.K.B. performed Western Blot experiments. V.E. performed RT-qPCRs and in vitro transcription experiments. E.N. conceived and supervised the project. B.M. and E.N. wrote the manuscript.

## Competing interests

The authors declare no competing interests.
