## [Peer Review File · Nature Communications]

Title: Pervasive Transcription-coupled DNA repair in *E. coli*REVIEWER COMMENTS

Reviewer #1 (Remarks to the Author):

Pervasive Transcription-coupled DNA Repair in E. coli

This important, carefully-performed paper presents striking data that challenge what has become the favored model for Transcription-Coupled Repair (TCR) in E. coli, while also redefining the relationship between GGR and TCR. The by-now canonical model for TCR, which has been significantly driven by in vitro experiments with purified proteins, holds that Mfd is both necessary and sufficient for this process. The central idea is that Mfd pushes the RNAP forward at a lesion site, thereby terminating transcription while recruiting the UvrAB to carry out NER. However, the genetic evidence for this model has always been less than compelling because an mfd mutant has, at best, a very marginal phenotype when exposed to UV light. Rather than the mfd phenotype being interpreted as in vivo data that challenges the in vitro model, instead the model based on in vitro data has been more or less presumed to be correct and the very modest mfd phenotype has been explained/hand-waved away by concluding that TCR is not very important in bacteria. This Mfd-dependent TCR model has remained pre-eminent despite a striking body of work, principally from the senior author's lab, which has offered strong support for an alternative model for Mfd-independent TCR. In this alternative model, the concerted action of UvrD and NusA in complex with RNAP promote the backtracking of RNAP stalled at a lesion. Mfd may contribute, but in a more modest manner and also by playing a different role than envisioned in the currently dominant TCR model. Also ppGpp promotes this process.

This paper, as well as the other co-submitted paper that was made available, is very important because the data and interpretations presented greatly expand our view of aspects of the complex physiological regulation of NER that has been traditionally interpreted as TCR, while at the same time integrating key previous findings into the new model. Rather than transcription-coupling being a process that exclusively favors the TS, this paper shows that transcription influences the NTS as well, with the difference between the TS and NTS repair being influenced by the strength of the transcription. A new concept suggested by the author's data is that UV-induced genotoxic stress promotes global antitermination and that this then influences where NER can occur.

The paper also highlights some physiological considerations that have not fully been considered because of the importance of in vitro data in driving the Mfd-dependent TCR model. For example, despite some familiarity with this topic, this reviewer was unaware that there are <100 Mfd molecules/cell with or without damage. Although there is no indication of where the accompanying paper has been submitted, but I feel that the complementary data contained within it definitely strengthens the authors' arguments in this manuscript.

I did not have any major concerns about the experiments in the paper. Rather, as is characteristic of papers from the Nudler lab, the experiments have been thought out with great care and are carefully performed. Numerous controls are included, for example checking that Rif does not diminish the protein

levels of UvrA, B, C or D.

I strongly recommend publication of this important manuscript.

Minor comments

1. The authors do a good job of rationalizing the seminal result of Mellon and Hanawalt and placing it in a new and larger context. However, less guidance is given concerning the interpretation or shortcomings of the papers from the Sancar lab (refs 23, 24) that argued that their data constituted *in vivo* evidence that Mfd is necessary and sufficient for TCR. A few more words here would be helpful.

2. For practical experimental reasons, the work described in this paper, as well as much of the other bacterial TCR research, has focused the removal of cyclobutane thymine dimers, which are recognized very well by the UvrABCD NER system. It might be worth pointing out that not all lesions that need to be repaired, for example N²-furfuryl-dG, are so easily recognized by UvrABCD NER. In such cases RNAP effectively serves an additional sensor for targeting the NER machinery to these harder-to-detect lesions. Drawing attention to this point could broaden the discussion and implications of this paper.

3. This may be irrelevant, but <https://doi.org/10.1128/jb.169.8.3435-3440.1987> refers to an antiterminator in the *uvrD* regulatory region. Given the key role of UvrD in this new model, is it possible that this could be important?

Reviewer #2 (Remarks to the Author):

Review of "Pervasive Transcription-coupled DNA repair in *E. coli*" by Martinez et al.

Overview:

Transcription-coupled repair (TCR) in bacteria has been well defined mechanistically including the components necessary and sufficient for TCR *in vitro* and *in vivo*. Central to the mechanism is the fact that RNAP is blocked by bulky DNA damage located in the transcribed strand (TS) but not the non-transcribed strand (NTS). Mfd protein binds to blocked RNAP, removes it and rapidly delivers the UvrA-UvrB proteins to the damage to mediate TCR which overall is faster than global genomic repair (GGR), which occurs in the absence of stalled RNAP. Repair is commonly complete about 30 minutes following damage; early in repair (to 10 min) there is considerable repair of the TS and later, following removal of TS damage, repair by GGR predominates. Previous reports from this laboratory claiming that UvrD (Epshtein et al. [2014] *Nature* 505, 372) and ppGpp (Kamarthapu et al. [2016] *Science* 352, 6288) are responsible for TCR in *E. coli* were found to be incorrect (Adebali et al. [2017] *Proc. Natl. Acad. Sci.* E2116-E2125; Adebali et al. [2017] *J. Biol. Chem.* 292, 18386).

In the submitted paper, measurements of RNA and DNA damage have been made to further investigate TCR and GGR and to test two hypotheses. The authors state first that since "transcription is more pervasive in bacteria than initially thought", "we hypothesize that NTS and intergenic regions may be

subject to TCR". In support of this they demonstrate antisense transcription which enhances repair in the NTS especially at the 3' end of genes. The level of antisense transcription is shown to be modulated by Rho, with more transcription and TCR when the transcription-termination function of Rho is reduced. Tangentially, Rho-dependent termination is shown to be reduced following DNA damage and antitermination is shown to contribute to overall transcription and repair. Secondly, the authors hypothesize that since "Mfd null cells display minimal sensitivity to UV and DNA damaging agents and overall rapid recovery from UV-induced damage occurs in the absence of Mfd19", then Mfd may not be "necessary and sufficient for all TCR that occurs in E. coli". To support this hypothesis, the authors report that rifampicin inhibits a substantial amount of repair including NTS repair, in line with their hypothesis that NTS repair may ensue from pervasive transcription in E. coli including widespread antisense transcription of annotated genes. Since rifampicin inhibits repair to a greater extent than the repair seen in mfd- cells, they conclude that the difference represents Mfd-independent TCR, and the authors further surmise that Mfd has only an indirect role in TCR. A model for Mfd-independent repair is proposed which includes a damage recognition complex containing RNAP and UvrA and UvrB. The repair pathway proposed supplants the currently recognized pathways for UvrA, UvrB and UvrC-mediated GGR and TCR mediated by Mfd and the Uvr proteins. The authors do not claim that the well-defined GGR and TCR reactions are in error, rather they claim that they simply don't occur in vivo.

Inspection of the manuscript reveals serious problems with method, originality, omissions, analyses and conclusions. A serious technical problem is that the method for measuring repair is low-resolution, and it appears from the data that it does not work well. As a result, quantitative comparisons of repair based upon this data cannot be made with confidence. There is a blatant failure to provide any direct evidence for the proposed, novel repair complex activity which supposedly supplants all 'conventional' GGR and TCR in vivo. Also, inhibition of repair by rifampicin treatment is not novel, it was described years ago (Lin, Kovalsky, Grossman [1998] *Nucleic Acids Res.* 26, 1466). That work made some serious effort to discover how the inhibition occurred, and the Grossman lab also discovered a complex consisting of RNAP and UvrA and UvrB (Lin, Kovalsky, Grossman [1997] *Nucleic Acids Res.* 25, 3151). This is essentially the complex proposed by Martinez et al. to catalyze all repair in E. coli. Martinez et al. fail to acknowledge this body of work and provide limited significant additional relevant information other than the inhibition is seen in repair of both strands. Already it is known that there is widespread antisense transcription and widespread non-annotated transcription in E. coli (Wade and Grainger [2014] *Nat. Rev. Microbiol.* 12, 647) that is associated with TCR in E. coli (Adebali et al. [2017] *Proc. Natl. Acad. Sci. USA* 114, E2116-E2125 and *J. Biol. Chem* 292, 18386). The demonstration by the authors that antisense and non-annotated transcription can lead to TCR is not surprising, it is incremental. The demonstration by the authors of antisense transcription and associated repair, as well as increased repair with antitermination is of a magnitude is too small to explain the overall rifampicin effect on NTS repair or to claim that all repair of the genome is transcription-coupled. The hypothesis mentions pervasive transcription, but while transcription is measured, pervasiveness is not shown and the action of rifampicin is not measured. Overall transcription in each strand is not compared with overall repair in each strand of active genes so as to explain the rifampicin effect and either identify or rule out GGR. In the experiments with rifampicin, in fact it appears from the author's data that one-third to one-fourth of total repair continues in the presence of rifampicin; however, the authors fail to discuss this important

point in relation to the claim that all repair is transcription-coupled. Furthermore, the rifampicin effect on repair has not been studied in detail, it may include direct and indirect effects on repair as considered by the Grossman lab but not here. The authors have not shown that repair of the NTS is exclusively due to antisense transcription, and that GGR does not contribute to NTS repair. The conclusion that there is no GGR in *E. coli* is not clearly stated, rather it is implied by the conclusion that all repair is TCR. Regarding the role of Mfd in repair, background information is mischaracterized and missing, the repair method provides data of poor quality, results are over-interpreted and conclusions overstated.

Specific comments:

In the title, Transcription-coupled DNA repair should be transcription-coupled DNA repair

The repair assay adapted for use here is inherently low resolution, similar to other assays that measure the amount of damage at zero time after UV and then measure damage at times thereafter, and calculate repair as the difference between the two time points. Since TCR occurs early in response to damage, this approach requires measurements of repair at early time points when the overall damage levels are the highest and the repair signal is the smallest, thus the repair signal is the small difference between two large numbers.

A related problem is sensitivity of detection and background. In the assay, before second adapter ligation, sonicated DNA is cleaved at sites of pyrimidine dimers with a cyclobutane pyrimidine dimer (CPD) specific nuclease, and the ends generated are prepared for ligation with the second adapter. At this point, any background DNA present with an end may ligate to the second adapter and potentially become a background signal. As control to characterize the repair signal produced by this procedure, Fig 1b shows the frequency of the four possible dipyrimidines (TT, TC, CT, CC) located in the recovered sequences at the positions next to the second adapter ligation site. Dipyrimidines are expected to map to this location because they are the targets for CPD formation and then the sites for CPD-specific nuclease digestion. If there is no background, then 100 percent of the sequence reads at this location should be dipyrimidines, anything less would suggest that the signal includes meaningless nonspecific background sequencing reads. From inspection of the top panel of Fig. 1b, it appears that before repair, at time zero (blue bars), the dipyrimidines constitute about 39 percent of the sequencing reads (15.5%TT+6%TC+11%CT+6.5%CC). This seems a very low value with an uncomfortably high background. With repair, the dipyridine levels at later time points decrease overall and therefore the background increases even further, based upon the data shown. It appears this experiment did not work.

Note there are two flaws, first the low total percent dipyrimidines recovered at zero time before repair, and secondly the decrease in the the total percent dipyrimidines at the different repair time points. While the number of CPDs should decrease with repair time, the percent detected should remain nearly constant if the experiment works.

The panel below suffers less from these problems. The total percent dipyrimidines at zero time (before repair) is calculated as 77.5%. The lower background makes sense in this case since in this panel a much higher dose was used and therefore there is a higher percentage of ends cleaved by the dimer specific nuclease to ligate to the second adapter. However, as repair proceeds, the total percent dipyrimidines does decrease and thus the background increases. Furthermore, this dataset suffers from the resolution problem described above, that is, at an early time point, when TCR is maximal, and with the high dose

required to obtain a signal, repair is measured as a small difference between two large numbers. This panel does not include the early time point (10 min) as is shown in the top panel, and most of the data sets shown in the paper using this method do not indicate the dose used.

Commonly a 20 minute repair time is used in the figures. At this time, TCR which peaks at early time points, is nearing completion, thus the TCR signal is relatively weak and TCR levels are under-represented in this paper. At later times in repair, after the TS is preferentially repaired, in fact, NTS repair should predominate. The authors are constrained by the deficiencies of the assay, low resolution at high doses/damage levels early in repair and limited detection with high background following the low dose and following repair (with diminution of the damage).

Figures such as 1c, d show percent decrease (PD) in TT-CPDs (20 min) using a y-axis in which there is no zero. Other figures 2a, c do show a zero on the y-axis. No explanation is found for the varied methods of analysis. Also, values such as 0.1, 0.3, 0.5... percent are given in these figures throughout. Presumably the authors mean 10, 30, 50... percent.

Regarding Figure 1c the paper states that “in the presence of Rif, both the TS and NTS are severely compromised in repair”, and later it is stated “Here, we provide evidence that transcription elongation is required for repair of lesions caused by UV genome-wide”. However, Figure 1c (and 5a) show substantial repair even when elongation is inhibited by rifampicin. In the presence of rifampicin, the figures show about 9 to 12 percent reduction in TTs. In the absence of rifampicin, the reduction is about 36 percent. Thus, about 9 to 12 divided by 36, or a third to a fourth of the total repair continues in the presence of rifampicin. This considerable repair seen in both strands in the presence of rifampicin appears to be attributable to GGR which by definition is transcription-independent. The authors provide no reasoning to justify the conclusion that GGR is absent *in vivo*. This conclusion, while not stated directly, is implied by their conclusion that all repair is via TCR.

Fig. 3a left panel is the first of several plots of this type. What is plotted is difficult to assess from the information available. Apparently it is the density of termination sites/termination events. It would be more informative to show termination events genome-wide scaled to an average gene such as in Fig. 2d and partially illustrated in the right panel of Fig. 3A. The right panel x-axis gives no bp or other indication of scale. Only a small portion of the signal is given for gene 3' end, however, it appears the readthrough is quite small in comparison here and in other figures. Similarly, the enhanced repair apparently due to antisense transcription shown in Fig. 2d, while in and of itself is interesting, it is quite small (48 to 44 average CPDs or 10%) and not directly linked to the rifampicin effect or quantitatively evaluated in relation to overall transcription and repair. The same lack of integration of interesting antitermination findings occurs with the antitermination due to UV data shown in Fig. 4. A ten percent effect is modest and of questionable significance, especially considering the method used.

It is unclear whether the fold change in *trxC* and *dsbB* are significant, Fig. 4c.

The second hypothesis tested is: since “Mfd null cells display minimal sensitivity to UV and DNA damaging agents and overall rapid recovery from UV-induced damage occurs in the absence of Mfd19”, then Mfd may not be “necessary and sufficient for all TCR that occurs in *E. coli*”. Reference 19 (“Mfd is required for rapid recovery of transcription following UV-induced DNA damage...” by Schawalow, B.J., Courcelle, C.T., Courcelle, J. [2012] *J. Bact.* 194, 2637) is mischaracterized by the authors. In fact, one of the phenotypes of Mfd is slow repair (George, Witkin [1974] *Molec.Gen.Genet.* 133, 283). This phenotype is consistent with TCR being characterized as a process of rapid repair of the TS of active

genes as compared to the NTS.

E. coli cells deficient in any of the Uvr proteins are highly sensitive to DNA damage because they do not repair the damage. Cells deficient in Mfd are not highly sensitive to DNA damage because they do repair the damage, albeit slowly. The modest UV sensitivity of *mfd*- cells is in fact consistent with MFD being “necessary and sufficient for all TCR that occurs in *E. coli*”.

In building their case, the authors not only misconstrue the information about Mfd and recovery from damage, they ignore other significant phenotypes regarding the roles of Mfd in mutagenesis. Mfd refers to the phenomenon mutation frequency decline (MFD), a phenotype so strong as to allow isolation of an *Mfd* mutant strain. In this case the target site for repair is in an anticodon-encoding loop of a suppressor tRNA, and repair leads to a decline in induced mutations. In addition to MFD, in wild type untreated cells, mutations are more prevalent in the presence of Mfd, and because of this Mfd has been considered a possible target for antibacterial drugs which may reduce mutagenesis and thus reduce development of drug resistance. Furthermore, *mfd*- cells demonstrate more damage-induced mutations in protein-encoding genes, and more importantly, mutations demonstrate a strand-specificity consistent with the known action of Mfd (Oller, Fijalkowska, Dunn, Schaaper [1992] *Proc.Natl.Acad.Sci.* 89, 11036). In wild type *E. coli*, in the *lacI* gene, 3.2 times more mutations were produced by DNA damage in the NTS compared to the TS. This is consistent with more repair of the TS by TCR. In *mfd*- cells, 4.5 times more mutations were produced by DNA damage in the TS compared to the NTS. This is consistent with the known inhibition of repair by RNAP stalled by damage in the TS. This same strandedness in mutation induction was observed in *B. subtilis*.

These phenotypes associated with Mfd including the modest UV sensitivity are entirely consistent with the body of evidence showing that Mfd is necessary and sufficient for TCR in *E. coli*.

In Fig. 5a, the authors show again the inhibition of repair by rifampicin and compare it to repair in wild type and *mfd*- cells at some undefined dose and repair time. The authors question the role of Mfd in repair; since *mfd* deletion does not reduce repair to the extent of inhibition by rifampicin, they question whether *mfd* is really responsible for TCR. Perhaps if accurately measured at early repair times, these relations would exhibit different proportions. In any event, the mechanism of TCR by Mfd is known. It has been characterized by numerous *in vitro* and *in vivo* investigations showing mechanistic details and the requirement for Mfd, including a method that measures repair genome-wide at single nucleotide resolution with high resolution (Adebali et al. [2017] *Proc. Natl. Acad. Sci.* 114, E2116; Adebali et al. [2017] *J. Biol. Chem* 292, 18386). What is not known is how does rifampicin produce the inhibition of repair that it does. In the prior study of rifampicin inhibition of repair (Lin et al., 1997), the extensive inhibition was attributed to the notion that transcription disrupts the nucleoid structure, making DNA more accessible for repair, so that with transcription inhibited by rifampicin, nucleoids inhibited repair. Thus there may be effects of rifampicin secondary to its effect on RNAP that produce the effects seen. It is inappropriate to discard something that has been characterized, the role of Mfd, based upon something incompletely characterized, the effects of rifampicin.

Fig. 5b shows inhibition of repair when Mfd is overexpressed. Though not cited or considered, purified Mfd added to *in vitro* repair reactions was previously shown to inhibit repair (Selby, Sancar [1991] *J.Biol.Chem* 270, 4890). Incidentally, addition of a low amount of UvrA stimulated repair, but higher amounts inhibited TCR. Mfd added to UvrA-repair-inhibited reactions caused a return of TCR. Thus, adding enzymes to complex reactions may lead to diverse outcomes. More may produce less in

biochemistry, and presumably there is a basis for regulation of gene expression and consequent levels of enzymes in cells. In light of this background the author's experimental approach is overly simplistic and the results do not add anything.

The authors suggest that the number of Mfd molecules per cell, <100, is insufficient to catalyse TCR. They re-propose the existence of a repair complex that includes RNAP, the damage recognition factor UvrA, and UvrB. By the author's logic, this complex is insufficient because there are only 16-20 UvrA molecules per cell (Sancar et al. [1981] *J.Mol. Biol.* 148, 45; Ghodke et al. [2020] *Nat. Commun.* 11, 1477).

Thus, the basis for their second hypothesis is mischaracterized and omits a great deal of important background information, and the hypothesis is therefore not well-founded, the mechanisms of rifampicin inhibition are not sufficiently well known as a basis to discard the role of Mfd in TCR, and the alternative mechanism involving a complex with RNAP, UvrA and UvrB, which has not been followed up in over 20 years, is a proposal with no direct evidence presented to support it.

The transcription-repair complex proposed by the authors also contains UvrD. UvrD is known to have an integral role in excision of DNA damage by the UvrABC(D) excision nuclease (Caron et al. [1985] *Proc. Natl. Acad. Sci.* 82, 4925; Husain et al. [1985] *Proc. Natl. Acad. Sci.* 82, 6774; Kumura et al. [1985] *Nucleic Acids Res.* 13, 1483; Orren et al. [1992] *J. Biol. Chem.* 267, 780; Adebali et al. [2017] *Proc. Natl. Acad. Sci.* 114, E2116-E2125). Following damage recognition and dual incision by UvrABC, UvrD helicase dissociates the Uvr proteins and the excised, damaged oligonucleotide from the genome (which is followed by repair synthesis and ligation). In the absence of UvrD, UvrABC make incisions nearly stoichiometric with UvrC and there is only very small turnover. Thus UvrD- cells are very sensitive to UV (though not as sensitive as UvrA, B or C mutants). In contrast, Mfd accelerates the transcribed strand repair but even in its absence the stalled RNAP is released by Rho and thus near-normal repair of both strands occurs, albeit slower than wild type in the TS. The authors seem unable to understand this simple and well-established fact and keep coming up with convoluted arguments as to why Mfd is not important for TCR, but UvrD is and continue to try various approaches to prove that "other investigator's Mfd" is only a minor factor in TCR and thus to have discovered the "True TCR Factor". Neither Mfd nor UvrD belong to a group of investigators or to anyone else. They belong to *E. coli* and other prokaryotes that have these proteins.

The authors state in the Abstract "Overall, our data suggests that GGR and TCR are essentially the same process required for complete repair of the bacterial genome." This is an obfuscation that appears to be based on the expectation that the general Abstract reader will not be clear about the definitions of TCR and GGR and will not realize that the authors really conclude that GGR and TCR as currently recognized do not occur in *E. coli* but are supplanted by a proposed pathway.

Reviewer #3 (Remarks to the Author):

The experiments are convincing that active transcription is important for NER in both the template strand and non-template strand. I am convinced as well that Mfd is not playing a crucial role in TCR. The experiments also do show that Mfd and Rho are somehow modulating TCR based transcription level of the gene. However, despite the authors' proposed model (which is pretty good), how this occurs is

unclear and will require further exploration. Nevertheless, this is a strong paper and seems appropriate for publication in Nat Comm. I have some minor comments that I think will strengthen the paper:

Active Transcription is Required for GGR, Figure 1; The literature discussed in the introduction does not conclusively demonstrate that Mfd is not important for NER processes. Despite the strong data in the paper, I think it is premature to exclude Mfd from the experiments described in the earlier parts of the paper, especially given its previously reported roles in recovery from transcription stalling stress (e.g., UV irradiation). The discussion of Mfd later in the paper is fine, but in the first set of experiments, I find its omission problematic.

The rifampicin treatment used to achieve a true “transcription off” condition seems particularly stringent—while experimentally necessary and controlled for in downstream analyses, I wonder if this treatment in combination with UV may be excessive. Please describe.

In Fig. 1c, “boxplots show the distribution of the percent decrease in TT-CPDs for each strand in gene bodies that had at least 1.5-fold TT-CPD enrichment at the 0-timepoint over NT-timepoint.” Please justify the 1.5-fold enrichment cutoff. This is repeated multiple times throughout the paper.

Figure 2; Please elaborate on how high, medium, and low transcription levels were assigned, or where this information was originally published. Are the bins contiguous, or are subset of representative genes used? This will affect the interpretation of the results. Fig. 2b is confusing. Upon close inspection, the trends the authors intend to illustrate are present, but it is difficult to critically analyze the data when figure contains so many different pieces.

NTS and Intergenic Regions are Subject to TCR; To determine which genes have an “antisense transcription preference,” the authors select a discrete region “in the last 100 bp and 50 bp downstream into intergenic regions.” I don’t have a major concern about these parameters, but because the group differences for the termination experiments are small, I would be interested to see if the effect is altered by expanding/ relaxing the cutoffs here. A justification of the region would also be appropriate.

Figure 3; Fig. 3b is poorly designed and does not clearly illustrate the points made in the manuscript. The authors should make their point in a different way. Additionally, please keep a consistent color scheme between all the figures; the red and orange in these plots is inconsistent.

Figure 4; In Figs. 4c, 4d, and 4e, a set of three genes is used to generalize the authors’ conclusions about UV-induced antitermination at Rho-dependent terminators. Please justify the selection of genes. The three included show consistent patterns with and without UV treatment but given the differences in magnitude in treatment effect I would like to see a few other terminators in the extended data.

Conclusions from each of these experiments would be strengthened with statistics. Absence of this analysis is particularly noticeable in the boxplots used repeatedly. With few exceptions, while the data do appear to show a trend, it is difficult to determine visually if this trend is significant.

Response to reviewers

We thank Reviewers 1 and 3 for their constructive feedback on our manuscript. The reviewers' comments and suggestions have enabled us to make textual revisions and experimental additions that clarify and strengthen our findings. As described below, we have revised the manuscript to address each of the points raised by the reviewers.

REVIEWER #1

This important, carefully-performed paper presents striking data that challenge what has become the favored model for Transcription-Coupled Repair (TCR in) *E. coli*, while also redefining the relationship between GGR and TCR. The by-now canonical model for TCR, which has been significantly driven by in vitro experiments with purified proteins, holds that Mfd is both necessary and sufficient for this process. The central idea is that Mfd pushes the RNAP forward at a lesion site, thereby terminating transcription while recruiting the UvrAB to carry out NER. However, the genetic evidence for this model has always been less than compelling because an *mfd* mutant has, at best, a very marginal phenotype when exposed to UV light. Rather than the *mfd* phenotype being interpreted as in vivo data that challenges the in vitro model, instead the model based on in vitro data has been more or less presumed to be correct and the very modest *mfd* phenotype has been explained/hand-waved away by concluding that TCR is not very important in bacteria. This Mfd-dependent TCR model has remain pre-eminent despite a striking body of work, principally from the senior author's lab, which has offered strong support for an alternative model for Mfd-independent TCR. In this alternative model, the concerted action of UvrD and NusA in complex with RNAP promote the backtracking of RNAP stalled a lesion. Mfd may contribute, but in more modest manner and also by playing a different role that envisioned in currently dominant TCR model. Also ppGpp promotes this process.

This paper, as well as the other co-submitted paper that was made available, is very important because the data and interpretations presented greatly expand our view of aspects of the complex physiological regulation of NER that has been traditionally interpreted as TCR, while at the same time integrating key previous findings into the new model. Rather than transcription-coupling being a process that exclusively favors the TS, this paper shows that transcription influences the NTS as well, with the difference between the TS and NTS repair being influenced by the strength of the transcription. An new concept suggested by the author's data is that UV-induced genotoxic stress promotes global antitermination and that that then influences where NER can occur.

The paper also highlights some physiological considerations that have not fully been considered because of the importance of in vitro data in driving the Mfd-dependent TCR model. For example, despite some familiarity with this topic, this reviewer was unaware that there are <100 Mfd molecules/cell with or without damage. Although there is no indication of where the accompanying paper has been submitted, but I feel that the complementary data contained within it definitely strengthens the authors' arguments in this manuscript.

I did not have any major concerns about the experiments in the paper. Rather, as is characteristic of papers from the Nudler lab, the experiments have been thought out with great care and are carefully performed. Numerous controls are included, for example checking that Rif does not diminish the protein levels of UvrA, B, C or D.

I strongly recommend publication of this important manuscript.

We thank the reviewer for their enthusiasm with our manuscript. We have taken into consideration their minor comments on how to further improve the clarity of the manuscript and have made additions to the text in response to their suggestions.

Minor comments

1. The authors do a good job of rationalizing the seminal result of Mellon and Hanawalt and placing it in a new and larger context. However, less guidance is given concerning the interpretation or shortcomings of the papers from the Sancar lab (refs 23, 24) that argued that their data constituted in vivo evidence that Mfd is necessary and sufficient for TCR. A few more words here would be helpful.

We agree that there were limitations from these studies referenced and thank the reviewer for reminding us to add them into our manuscript. **We have added these shortcomings in the introduction.**

2, For practical experimental reasons, the work described in this paper, as well as much of the other bacterial TCR research, has focused the removal of cyclobutane thymine dimers, which are recognized very well by the UvrABCD NER system. It might be worth pointing out that not all lesions that need to be repaired, for example N2-furfuryl-dG, are so easily recognized by UvrABCD NER. In such cases RNAP effectively serves an additional sensor for targeting the NER machinery to these harder-to-detect lesions. Drawing attention to this point could broaden the discussion and implications of this paper.

We appreciated the reviewer for bringing up this interesting point and agree that RNAP scanning would serve as an effective way to measure lesions beyond CPDs. **We have added this point to our discussion section.**

3. This may be irrelevant, but <https://doi.org/10.1128/jb.169.8.3435-3440.1987> refers to an antiterminator in the uvrD regulatory region. Given the key role of UvrD in this new model, is it possible that this could be important?

This is an interesting study on the transcriptional regulation of UvrD and the authors of this study find that mutating a termination loop within the uvrD promoter results in increased transcription at this promoter. We focused on Rho-dependent termination in our study and did not confirm that UV can increase readthrough past intrinsic terminators globally. And because the authors here had to mutate the terminator before seeing increased transcription (only after UV), we expect that this would not occur in our experimental setup. However, it would be interesting to perform CPD-seq on a strain with these mutations to determine if the increased UvrD that results from these mutations has an effect on global NER.

Reviewer #2 (Remarks to the Author):

Review of “Pervasive Transcription-coupled DNA repair in *E. coli*” by Martinez et al.

Overview:

Transcription-coupled repair (TCR) in bacteria has been well defined mechanistically including the components necessary and sufficient for TCR in vitro and in vivo. Central to the mechanism is the fact that RNAP is blocked by bulky DNA damage located in the transcribed strand (TS) but not the non-transcribed strand (NTS). Mfd protein binds to blocked RNAP, removes it and rapidly delivers the UvrA-UvrB proteins to the damage to mediate TCR which overall is faster than global genomic repair (GGR), which occurs in the absence of stalled RNAP. Repair is commonly complete about 30 minutes following damage; early in repair (to 10 min) there is considerable repair of the TS and later, following removal of TS damage, repair by GGR predominates. Previous reports from this laboratory claiming that UvrD (Epshtein et al. [2014] *Nature* 505, 372) and ppGpp (Kamarthapu et al. [2016] *Science* 352, 6288) are responsible for TCR in *E. coli* were found to be incorrect (Adebali et al. [2017] *Proc. Natl. Acad. Sci.* E2116-E2125; Adebali et al. [2017] *J. Biol. Chem.* 292, 18386).

We disagree with the claim that the previous publications from our lab have been proven wrong by the two papers cited by this reviewer. There is a *fundamental* flaw in the analysis from these studies. First, the authors base their claim that Mfd is the sole TCR factor by comparing the TS/NTS ratio WT vs *mfd*(-) strain. They observe that deleting Mfd causes the TS/NTS ratio to decrease. However, a decrease in the TS/NTS can be caused by a decrease in TS repair or an increase in NTS repair. The authors failed to analyze the repair of each strand separately and therefore are not properly assessing TCR. Our manuscript shows that transcription is necessary for repair of not only the TS, but also the NTS. Further, analysis of the repair of each strand shows that Mfd contributes to the repair of the TS of highly transcribed genes. However, we see that Mfd is not required for the majority of TCR that occurs in TS of lower transcribed genes and the NTS. We even show that Mfd's role as a termination factor inhibits TCR in the NTS.

The authors performed two experiments from these studies to examine the role of UvrD in repair. First they did XR-seq in a *uvrD*(-) strain and saw that this strain did not change the TS/NTS repair ratio compared to WT. However, as mentioned above, our results demonstrate that TS/NTS ratio is an inadequate parameter to assess TCR, as both strands must be transcribed for NER to occur. Because UvrD is essential for TCR on both strands, one does not expect TS/NTS ratio to change significantly. Indeed, our CPD-seq data shows that deletion of UvrD causes a severe deficiency in repair of both the TS and NTS and that the TS repair preference is not as strong as in WT cells. Yet, we do observe a very low TS repair preference and this is because a low amount of TCR can occur in the absence of UvrD due to spontaneous backtracking.

Furthermore, the *uvrD*(-) strain produced a major artifact using XR-seq where the excised oligonucleotide accumulates more than in all the other strains. Therefore, it is difficult to compare a *uvrD*(-) strain with any other strain using XR-seq.

Again, our paper shows that TCR occurs on both strands. Any areas with high RNAP presence will have faster repair than lower transcribed regions such as the NTS. Thus, the results presented in the present paper found Adebali et al conclusions to be fundamentally incorrect. We also note that Adebali et al simply disregarded all the biochemical, genetic, and functional

data we presented in our previous publications, which are perfectly consistent with the conclusions reached in our present manuscript and the one we co-submitted (Bharati et al.).

In the submitted paper, measurements of RNA and DNA damage have been made to further investigate TCR and GGR and to test two hypotheses. The authors state first that since “transcription is more pervasive in bacteria than initially thought”, “we hypothesize that NTS and intergenic regions may be subject to TCR”. In support of this they demonstrate antisense transcription which enhances repair in the NTS especially at the 3’ end of genes. The level of antisense transcription is shown to be modulated by Rho, with more transcription and TCR when the transcription-termination function of Rho is reduced. Tangentially, Rho-dependent termination is shown to be reduced following DNA damage and antitermination is shown to contribute to overall transcription and repair. Secondly, the authors hypothesize that since “Mfd null cells display minimal sensitivity to UV and DNA damaging agents and overall rapid recovery from UV-induced damage occurs in the absence of Mfd19”, then Mfd may not be “necessary and sufficient for all TCR that occurs in *E. coli*”. To support this hypothesis, the authors report that rifampicin inhibits a substantial amount of repair including NTS repair, in line with their hypothesis that NTS repair may ensue from pervasive transcription in *E. coli* including widespread antisense transcription of annotated genes. Since rifampicin inhibits repair to a greater extent than the repair seen in *mfd*- cells, they conclude that the difference represents Mfd-independent TCR, and the authors further surmise that Mfd has only an indirect role in TCR. A model for Mfd-independent repair is proposed which includes a damage recognition complex containing RNAP and UvrA and UvrB. The repair pathway proposed supplants the currently recognized pathways for UvrA, UvrB and UvrC-mediated GGR and TCR mediated by Mfd and the Uvr proteins. The authors do not claim that the well-defined GGR and TCR reactions are in error, rather they claim that they simply don’t occur *in vivo*.

Inspection of the manuscript reveals serious problems with method, originality, omissions, analyses and conclusions. A serious technical problem is that the method for measuring repair is low-resolution, and it appears from the data that it does not work well. As a result, quantitative comparisons of repair based upon this data cannot be made with confidence. There is a blatant failure to provide any direct evidence for the proposed, novel repair complex activity which supposedly supplants all ‘conventional’ GGR and TCR *in vivo*. Also, inhibition of repair by rifampicin treatment is not novel, it was described years ago (Lin, Kovalsky, Grossman [1998] *Nucleic Acids Res.* 26, 1466). That work made some serious effort to discover how the inhibition occurred, and the Grossman lab also discovered a complex consisting of RNAP and UvrA and UvrB (Lin, Kovalsky, Grossman [1997] *Nucleic Acids Res.* 25, 3151). This is essentially the complex proposed by Martinez et al. to catalyze all repair in *E. coli*. Martinez et al. fail to acknowledge this body of work and provide limited significant additional relevant information other than the inhibition is seen in repair of both strands. Already it is known that there is widespread antisense transcription and widespread non-annotated transcription in *E. coli* (Wade and Grainger [2014] *Nat. Rev. Microbiol.* 12, 647) that is associated with TCR in *E. coli* (Adebali et al. [2017] *Proc. Natl. Acad. Sci. USA* 114, E2116-E2125 and *J. Biol. Chem* 292, 18386). The demonstration by the authors that antisense and non-annotated transcription can lead to TCR is not surprising, it is incremental. The demonstration by the authors of antisense transcription and associated

repair, as well as increased repair with antitermination is of a magnitude is too small to explain the overall rifampicin effect on NTS repair or to claim that all repair of the genome is transcription-coupled. The hypothesis mentions pervasive transcription, but while transcription is measured, pervasiveness is not shown and the action of rifampicin is not measured. Overall transcription in each strand is not compared with overall repair in each strand of active genes so as to explain the rifampicin effect and either identify or rule out GGR. In the experiments with rifampicin, in fact it appears from the author's data that one-third to one-fourth of total repair continues in the presence of rifampicin; however, the authors fail to discuss this important point in relation to the claim that all repair is transcription-coupled. Furthermore, the rifampicin effect on repair has not been studied in detail, it may include direct and indirect effects on repair as considered by the Grossman lab but not here. The authors have not shown that repair of the NTS is exclusively due to antisense transcription, and that GGR does not contribute to NTS repair. The conclusion that there is no GGR in *E. coli* is not clearly stated, rather it is implied by the conclusion that all repair is TCR. Regarding the role of Mfd in repair, background information is mischaracterized and missing, the repair method provides data of poor quality, results are over-interpreted and conclusions overstated.

Specific comments:

In the title, Transcription-coupled DNA repair should be transcription-coupled DNA repair. The repair assay adapted for use here is inherently low resolution, similar to other assays that measure the amount of damage at zero time after UV and then measure damage at times thereafter, and calculate repair as the difference between the two time points. Since TCR occurs early in response to damage, this approach requires measurements of repair at early time points when the overall damage levels are the highest and the repair signal is the smallest, thus the repair signal is the small difference between two large numbers. A related problem is sensitivity of detection and background. In the assay, before second adapter ligation, sonicated DNA is cleaved at sites of pyrimidine dimers with a cyclobutane pyrimidine dimer (CPD) specific nuclease, and the ends generated are prepared for ligation with the second adapter. At this point, any background DNA present with an end may ligate to the second adapter and potentially become a background signal. As control to characterize the repair signal produced by this procedure, Fig 1b shows the frequency of the four possible dipyrimidines (TT, TC, CT, CC) located in the recovered sequences at the positions next to the second adapter ligation site. Dipyrimidines are expected to map to this location because they are the targets for CPD formation and then the sites for CPD-specific nuclease digestion. If there is no background, then 100 percent of the sequence reads at this location should be dipyrimidines, anything less would suggest that the signal includes meaningless nonspecific background sequencing reads. From inspection of the top panel of Fig. 1b, it appears that before repair, at time zero (blue bars), the dipyrimidines constitute about 39 percent of the sequencing reads (15.5%TT+6%TC+11%CT+6.5%CC). This seems a very low value with an uncomfortably high background.

The CPD-seq assay is not low resolution as it is able to sequence the specific site where damage is cleaved by T4 PDG, and, hence, it is a single-nucleotide-resolution method. This assay has been shown to correctly identify sites of UV damage in a published paper that was studying CPD damage in yeast¹. We have modified this assay for Illumina sequencing and also

adapted it to study DNA repair in *E. coli*. The protocol specifically enriches for sites that were cleaved by specific enzymes by adding a biotinylated adapter to the cleavage site that can be pulled down with streptavidin. In addition, there is a 3' end blocking step that specifically reduces unwanted background by using terminal transferase and a terminal nucleotide to block any nonspecific 3' adapter ligation. The previous published paper and our paper show that some background occurs most likely because we treat the samples with an AP endonuclease that also cleave abasic sites. Regardless of background, our method shows an enrichment for CPD lesions, with TT being the most common dinucleotide sequence that is adjacent to a cleavage site in UV treated samples. We also show that this enrichment clearly increases with increased UV dose. Further, to avoid any noise in our data, we limit all of our downstream analyses to only reads that were adjacent to TT sites because those are the most enriched reads immediately after UV damage exposure. Even if 15.5% of the reads are adjacent to TT, it still amounts to sufficient coverage of the *E. coli* genome by only using these reads because we sequenced at an extremely high depth (mean depth > 16 million reads per sample) and could therefore afford to limit our analyses to only these reads. It is clear from our data that TCR can be measured and we show that many genes with high transcription level have the highest TS repair preference. This observation has been seen in the literature since the discovery of TCR and the fact that our data also show this, proves that our method is working. It is common for NGS assays to have background, in fact, the XR-seq assay the reviewer cites has a significant amount of background and a majority of reads need to be filtered from these analyses as well^{2,3}.

With repair, the dipyrimidine levels at later time points decrease overall and therefore the background increases even further, based upon the data shown. It appears this experiment did not work.

We expect there to be **less** dipyrimidine enrichment during recovery because they are being repaired, we argue that this result shows the experiment does work. This was also shown in the previously published CPD-seq paper¹.

Note there are two flaws, first the low total percent dipyrimidines recovered at zero time before repair, and secondly the decrease in the the total percent dipyrimidines at the different repair time points. While the number of CPDs should decrease with repair time, the percent detected should remain nearly constant if the experiment works. The panel below suffers less from these problems. The total percent dipyrimidines at zero time (before repair) is calculated as 77.5%. The lower background makes sense in this case since in this panel a much higher dose was used and therefore there is a higher percentage of ends cleaved by the dimer specific nuclease to ligate to the second adapter. However, as repair proceeds, the total percent dipyrimidines does decrease and thus the background increases.

We also expect the background to increase when there is less damage because less cleavage by T4 PDG will occur. We do not understand how this result is incorrect or leads to any flaws in the analysis. The point the reviewer is trying to make in this paragraph is not clear. Once again, this same result was observed in the previously published study where this assay was used.

Furthermore, this dataset suffers from the resolution problem described above, that is, at an early time point, when TCR is maximal, and with the high dose required to obtain a signal, repair is measured as a small difference between two large numbers. This panel

does not include the early time point (10 min) as is shown in the top panel, and most of the data sets shown in the paper using this method do not indicate the dose used.

Commonly a 20 minute repair time is used in the figures. At this time, TCR which peaks at early time points, is nearing completion, thus the TCR signal is relatively weak and TCR levels are under-represented in this paper. At later times in repair, after the TS is preferentially repaired, in fact, NTS repair should predominate. The authors are constrained by the deficiencies of the assay, low resolution at high doses/damage levels early in repair and limited detection with high background following the low dose and following repair (with diminution of the damage).

The reviewer keeps on referring to the CPD-seq assay as low resolution. We explain above why it is not, but also would like to add that this assay has just as high of resolution as the XR-seq studies they cite, but also has the advantage of being able to measure **several** recovery timepoints from the **same** sample whereas this is not possible using XR-seq. It is clear from our data (Fig 2a) that the **strongest** preference in TS repair in highly transcribed genes **occurs at the 20-minute timepoint**. Therefore, TCR is not underrepresented in this timepoint. Even our Mfd experiment shows that deleting Mfd at this timepoint results in a deficiency in this repair preference. We just have a different model for the role of Mfd in repair.

Figures such as 1c, d show percent decrease (PD) in TT-CPDs (20 min) using a y-axis in which there is no zero. Other figures 2a, c do show a zero on the y-axis. No explanation is found for the varied methods of analysis. Also, values such as 0.1, 0.3, 0.5... percent are given in these figures throughout. Presumably the authors mean 10, 30, 50... percent.

The zero is in all of the y-axes, however depending on the labels of the figure the zero may not show, so it is unclear what the problem is that the reviewer is referring to. Also, these values are represented in frequencies rather than percentages, but that does not change the overall results we present.

Regarding Figure 1c the paper states that “in the presence of Rif, both the TS and NTS are severely compromised in repair”, and later it is stated “Here, we provide evidence that transcription elongation is required for repair of lesions caused by UV genome-wide”. However, Figure 1c (and 5a) show substantial repair even when elongation is inhibited by rifampicin. In the presence of rifampicin, the figures show about 9 to 12 percent reduction in TTs. In the absence of rifampicin, the reduction is about 36 percent. Thus, about 9 to 12 divided by 36, or a third to a fourth of the total repair continues in the presence of rifampicin. This considerable repair seen in both strands in the presence of rifampicin appears to be attributable to GGR which by definition is transcription-independent. The authors provide no reasoning to justify the conclusion that GGR is absent *in vivo*. This conclusion, while not stated directly, is implied by their conclusion that all repair is via TCR.

We do not agree that a substantial amount of repair still occurs in the rif conditions we used. We do agree that some *residual* recovery is observed in our rif conditions especially compared to a *uvra(-)* strain. We hypothesize that GGR either has little contribution to NER or is absent based on the overall data from this manuscript and the complimentary manuscript from our lab (Bharati et al. co-submitted). We also cannot be certain that the rif conditions we used were able to stop

all active RNAPs. It is likely that a small amount of residual transcription, especially TCRC, which are more resistant to termination, can lead to repair. In our complimentary manuscript we needed to create a special “insulator” strain where we inserted intrinsic terminator sequences on each side of a gene to guarantee that no RNAPs can enter this region. We find that repair no longer occurs on either strand in this insulated region (in Bharati et al. co-submitted, Fig. 5, Extended Data Figs. 11 and 12).

Fig. 3a left panel is the first of several plots of this type. What is plotted is difficult to assess from the information available. Apparently it is the density of termination sites/termination events. It would be more informative to show termination events genome-wide scaled to an average gene such as in Fig. 2d and partially illustrated in the right panel of Fig. 3A. The right panel x-axis gives no bp or other indication of scale. Only a small portion of the signal is given for gene 3' end, however, it appears the readthrough is quite small in comparison here and in other figures.

We agree that the readthrough is mild, which is why it is so interesting that even this mild readthrough alters repair genome wide. The bigger readthrough is expected to be toxic to the cell, as Rho termination is essential for viability⁴. The figures are a meta-analysis and therefore they do represent an average of defined Rho termination sites⁵. The methods describe how these plots were generated and the window length.

Similarly, the enhanced repair apparently due to antisense transcription shown in Fig. 2d, while in and of itself is interesting, it is quite small (48 to 44 average CPDs or 10%) and not directly linked to the rifampicin effect or quantitatively evaluated in relation to overall transcription and repair.

This analysis is a metanalysis used to demonstrate a trend across many genes in the analysis. Compared to the TS, the NTS clearly has a bias in repair at the gene end. This was also shown in several other gene windows.

The same lack of integration of interesting antitermination findings occurs with the antitermination due to UV data shown in Fig. 4. A ten percent effect is modest and of questionable significance, especially considering the method used.

It is unclear what “ten percent effect” the reviewer is referring to from this figure.

It is unclear whether the fold change in *trxC* and *dsbB* are significant, Fig. 4c.

The p-values were calculated for *proA*, *trxC* and *dsbB*. The *proA* and *dsbB* terminators show a significant increase in readthrough ($p < 0.05$). The *trxC* had a p-value close to this threshold at $p = 0.06$. However, the qPCR experiments combined with the genome-wide data show strong evidence for global transcription readthrough after UV.

The second hypothesis tested is: since “Mfd null cells display minimal sensitivity to UV and DNA damaging agents and overall rapid recovery from UV-induced damage occurs in the absence of Mfd19”, then Mfd may not be “necessary and sufficient for all TCR that occurs in *E. coli*”. Reference 19 (“Mfd is required for rapid recovery of transcription following UV-

induced DNA damage...” by Schallow, B.J., Courcelle, C.T., Courcelle, J. [2012] J. Bact. 194, 2637) is mischaracterized by the authors. In fact, one of the phenotypes of Mfd is slow repair (George, Witkin [1974] Molec.Gen.Genet. 133, 283). This phenotype is consistent with TCR being characterized as a process of rapid repair of the TS of active genes as compared to the NTS.

While it does seem that Mfd delays the repair of the TS, the repair of the overall genome is not severely affected in *mfd(-)* cells. Our own experiments (Bharati et al co-submitted, Extended Data Fig. 8a) show similar results as the Schallow et al. paper⁶.

E. coli cells deficient in any of the Uvr proteins are highly sensitive to DNA damage because they do not repair the damage. Cells deficient in Mfd are not highly sensitive to DNA damage because they do repair the damage, albeit slowly. The modest UV sensitivity of *mfd-* cells is in fact consistent with MFD being “necessary and sufficient for all TCR that occurs in *E. coli*”.

In building their case, the authors not only misconstrue the information about Mfd and recovery from damage, they ignore other significant phenotypes regarding the roles of Mfd in mutagenesis. Mfd refers to the phenomenon mutation frequency decline (MFD), a phenotype so strong as to allow isolation of an Mfd mutant strain. In this case the target site for repair is in an anticodon-encoding loop of a suppressor tRNA, and repair leads to a decline in induced mutations. In addition to MFD, in wild type untreated cells, mutations are more prevalent in the presence of Mfd, and because of this Mfd has been considered a possible target for antibacterial drugs which may reduce mutagenesis and thus reduce development of drug resistance. Furthermore, *mfd-* cells demonstrate more damage-induced mutations in protein-encoding genes, and more importantly, mutations demonstrate a strand-specificity consistent with the known action of Mfd (Oller, Fijalkowska, Dunn, Schaaper [1992] Proc.Natl.Acad.Sci. 89, 11036). In wild type *E. coli*, in the *lacI* gene, 3.2 times more mutations were produced by DNA damage in the NTS compared to the TS. This is consistent with more repair of the TS by TCR. In *mfd-* cells, 4.5 times more mutations were produced by DNA damage in the TS compared to the NTS. This is consistent with the known inhibition of repair by RNAP stalled by damage in the TS. This same strandedness in mutation induction was observed in *B. subtilis*.

These phenotypes associated with Mfd including the modest UV sensitivity are entirely consistent with the body of evidence showing that Mfd is necessary and sufficient for TCR in *E. coli*.

Our model does not contradict these phenotypes associated with Mfd. Our data shows that Mfd would be important for the rapid repair of the TS of highly transcribed genes, such as the TS of the *lacI* gene in the experiment described by the reviewer. However, we propose that Mfd is important for recovery of the TS *indirectly* rather than directly acting as a TCR factor. Mfd would play a role in removing elongation complex “traffic” in highly transcribed regions so that the TCRC complex can access the lesion site (Extended Data Fig. 9 and Bharati et al., Ext Data Fig. 16). However, in regions that are not highly transcribed, such as the NTS, we find that Mfd inhibits repair through its role as a termination factor (similar to Rho). In fact, our model explains how Mfd can be anti-mutagenic in certain contexts (such as the TS of highly transcribed genes) and pro-mutagenic in other contexts. Mfd has been

found to be pro-mutagenic⁷ and this results aligns with our data showing that Mfd can inhibit NTS repair. Mfd has very modest UV sensitivity because it only modulates NER, mildly improving repair at highly transcribed regions and mildly inhibiting repair at lowly transcribed regions. The model of Mfd as a cleanup factor during repair is more consistent with this modest UV sensitivity because this termination function can be replaced by Rho in the absence of Mfd.

In Fig. 5a, the authors show again the inhibition of repair by rifampicin and compare it to repair in wild type and *mfd*- cells at some undefined dose and repair time. The authors question the role of Mfd in repair; since *mfd* deletion does not reduce repair to the extent of inhibition by rifampicin, they question whether *mfd* is really responsible for TCR. Perhaps if accurately measured at early repair times, these relations would exhibit different proportions. In any event, the mechanism of TCR by Mfd is known. It has been characterized by numerous *in vitro* and *in vivo* investigations showing mechanistic details and the requirement for Mfd, including a method that measures repair genome-wide at single nucleotide resolution with high resolution (Adebali et al. [2017] Proc. Natl. Acad. Sci. 114, E2116; Adebali et al. [2017] J. Biol. Chem 292, 18386). What is not known is how does rifampicin produce the inhibition of repair that it does. In the prior study of rifampicin inhibition of repair (Lin et al., 1997), the extensive inhibition was attributed to the notion that transcription disrupts the nucleoid structure, making DNA more accessible for repair, so that with transcription inhibited by rifampicin, nucleoids inhibited repair. Thus there may be effects of rifampicin secondary to its effect on RNAP that produce the effects seen. It is inappropriate to discard something that has been characterized, the role of Mfd, based upon something incompletely characterized, the effects of rifampicin.

We are now aware of this paper from Lin et al, however we argue that this paper supports our model that active transcription is necessary for repair. This paper shows that rif can inhibit most, if not all, NER *ex vivo* in the “nucleoid” fraction and proposes that RNAP can bind UvrA directly, which is also what we describe in our complimentary manuscript (Bharati et al.). The authors of this paper also propose that most NER is likely to be coupled to transcription and that Mfd does not play a significant role in TCR. However, this paper did not provide proper controls to show that the rif results are not from secondary effects. In our manuscript we show through a Western Blot that all of the factors required for NER (UvrA, UvrB, UvrC, UvrD) are present in the same amounts after our temporary Rif exposure. There is no evidence that disruption of the nucleoid inhibits repair and therefore the reviewers claim that the rif effect is due to this secondary effect is unsubstantiated. In addition it has been shown after rif exposure the nucleoid stays intact⁸ and therefore the inhibition of repair is not caused by this imaginary “secondary effect”.

In addition to the rif experiment we show in this manuscript, we also performed an experiment in the co-submitted manuscript that supports the claim that repair cannot occur in the absence of transcription. In this experiment, we created an *E. coli* strain with a genomic region that is insulated by six intrinsic terminators (Bharati et al, Fig 5f). Using this insulator technology, we are able to create a genomic region that does not contain active RNAP without inhibiting global transcription and therefore secondary effects from

rif are not possible. In this experiment we also observe that repair cannot occur in the absence of transcription (Bharati et al, Fig f-i).

Fig. 5b shows inhibition of repair when Mfd is overexpressed. Though not cited or considered, purified Mfd added to *in vitro* repair reactions was previously shown to inhibit repair (Selby, Sancar [1991] J.Biol.Chem 270, 4890). Incidentally, addition of a low amount of UvrA stimulated repair, but higher amounts inhibited TCR. Mfd added to UvrA-repair-inhibited reactions caused a return of TCR. Thus, adding enzymes to complex reactions may lead to diverse outcomes. More may produce less in biochemistry, and presumably there is a basis for regulation of gene expression and consequent levels of enzymes in cells. In light of this background the author's experimental approach is overly simplistic and the results do not add anything.

Mfd was first characterized as a TCR factor *in vitro* using reactions that had excess Mfd, the claim that "more may produce less" can also be said for these early experiments. According to the traditional model of TCR, excess Mfd *in vivo* should enhance repair because Mfd would then recruit repair proteins to even more RNAPs. However, we do not observe this result. Excess Mfd inhibits repair *in vivo*.

The authors suggest that the number of Mfd molecules per cell, <100, is insufficient to catalyse TCR. They re-propose the existence of a repair complex that includes RNAP, the damage recognition factor UvrA, and UvrB. By the author's logic, this complex is insufficient because there are only 16-20 UvrA molecules per cell (Sancar et al. [1981] J.Mol. Biol. 148, 45; Ghodke et al. [2020] Nat. Commun. 11, 1477).

Thus, the basis for their second hypothesis is mischaracterized and omits a great deal of important background information, and the hypothesis is therefore not well-founded, the mechanisms of rifampicin inhibition are not sufficiently well known as a basis to discard the role of Mfd in TCR, and the alternative mechanism involving a complex with RNAP, UvrA and UvrB, which has not been followed up in over 20 years, is a proposal with no direct evidence presented to support it.

Our complimentary paper (Bharati et al. co-submitted) provides *direct* *in vivo* and *in vitro* evidence for the RNAP complex with UvrA/B/D. Using quantitative mass spectrometry (qMS), we estimate that the number of UvrA molecules is in the low hundreds per cell prior to genotoxic stress and that >30% of those molecules interact with elongating RNAP at any given moment (Extended Data Fig. 2b in Bharati et al co-submitted). We agree that it was somewhat misleading to argue for a very low number of Mfd molecules citing Ho et al, 2018 and Schmidt et al., 2016 (they detected <50 per E. coli cell^{6,9}), whereas our own estimate based on qMS indicates a larger number of Mfd molecules per cell (~150-200). We have edited our discussion to reflect this new result, but the main point of the discussion does not change. Mfd can interact with RNAP regardless of DNA damage^{9,10} and therefore the chances it can be recruited to the small fraction of RNAPs stalled exactly at the site of damage *in vivo* are negligible. According to the traditional model, Mfd *somehow* very rapidly finds only those RNAP molecules *in vivo* that were stalled at DNA damage sites, ignoring the vast majority of others with whom Mfd can interact with the same efficiency. Then, Mfd *somehow* remains at the lesion site long enough to bring UvrA there. All these critical assumptions of the traditional Mfd model have never been directly supported by any biochemical or *in vivo* data.

The pre-TCRC/TCRC model we describe in this and Bharati et al. manuscripts proposes that even a small number of UvrA molecules (relative to RNAPs) should be sufficient for TCR as UvrA molecules would be continuously present as a part of the pre-TCRC/TCRC complex which scans the genome in one dimension for rapid lesion detection. The unbiased XLMS-driven structural models of the pre-TCRC/TCRC described in the co-submitted manuscript are consistent with the available structural models of UvrAB-DNA complexes.

The transcription-repair complex proposed by the authors also contains UvrD. UvrD is known to have an integral role in excision of DNA damage by the UvrABC(D) excision nuclease (Caron et al. [1985] Proc. Natl. Acad. Sci. 82, 4925; Husain et al. [1985] Proc. Natl. Acad. Sci. 82, 6774; Kumura et al. [1985] Nucleic Acids Res. 13, 1483; Orren et al. [1992] J. Biol. Chem. 267, 780; Adebali et al. [2017] Proc. Natl. Acad. Sci. 114, E2116-E2125). Following damage recognition and dual incision by UvrABC, UvrD helicase dissociates the Uvr proteins and the excised, damaged oligonucleotide from the genome (which is followed by repair synthesis and ligation). In the absence of UvrD, UvrABC make incisions nearly stoichiometric with UvrC and there is only very small turnover. Thus UvrD- cells are very sensitive to UV (though not as sensitive as UvrA, B or C mutants). In contrast, Mfd accelerates the transcribed strand repair but even in its absence the stalled RNAP is released by Rho and thus near-normal repair of both strands occurs, albeit slower than wild type in the TS. The authors seem unable to understand this simple and well-established fact and keep coming up with convoluted arguments as to why Mfd is not important for TCR, but UvrD is and continue to try various approaches to prove that “other investigator’s Mfd” is only a minor factor in TCR and thus to have discovered the “True TCR Factor”. Neither Mfd nor UvrD belong to a group of investigators or to anyone else. They belong to E. coli and other prokaryotes that have these proteins.

We agree that these factors do not “belong” to anyone. In fact, the main point of our paper does not focus on either of these two factors. The main finding that we emphasize in our manuscript is that transcription is pervasive and RNAP acts as a surveyor of DNA damage across the entire genome. Because transcription is not limited to only the TS, TCR would occur in other genomic regions such as the NTS and intergenic regions. Despite the reviewer claiming that this study is lacking in novelty, a global assessment of TCR in these genomic regions had not been performed before the submission of our manuscript. The XR-seq studies that the reviewer cites did not focus on this question and mainly focused on trying to prove that Mfd was the only factor that can be involved in TCR. Our manuscript, as well as the co-submitted manuscript (Bharati et al), show that TCR is in fact an essential pathway required for NER. We do not ignore Mfd and explain how this factor plays a role in the pervasive TCR model we describe. It is ironic the reviewer questions the novelty of our work, as it completely changes the very concepts the reviewer was basing all their arguments.

The authors state in the Abstract “Overall, our data suggests that GGR and TCR are essentially the same process required for complete repair of the bacterial genome.” This is an obfuscation that appears to be based on the expectation that the general Abstract reader will not be clear about the definitions of TCR and GGR and will not realize that the authors really conclude that GGR and TCR as currently recognized do not occur in E. coli but are supplanted by a proposed pathway.

1. Mao, P., Smerdon, M. J., Roberts, S. A. & Wyrick, J. J. Chromosomal landscape of UV damage formation and repair at single-nucleotide resolution. *Proc Natl Acad Sci USA* **113**, 9057 (2016).
2. Adebali, O., Chiou, Y.-Y., Hu, J., Sancar, A. & Selby, C. P. Genome-wide transcription-coupled repair in Escherichia coli is mediated by the Mfd translocase. *Proc Natl Acad Sci USA* **114**, E2116 (2017).
3. Adebali, O., Sancar, A. & Selby, C. P. Mfd translocase is necessary and sufficient for transcription-coupled repair in Escherichia coli. *J Biol Chem* **292**, 18386–18391 (2017).
4. Cardinale, C. J. *et al.* Termination factor Rho and its cofactors NusA and NusG silence foreign DNA in E. coli. *Science* **320**, 935–938 (2008).
5. Dar, D. & Sorek, R. High-resolution RNA 3'-ends mapping of bacterial Rho-dependent transcripts. *Nucleic Acids Research* **46**, 6797–6805 (2018).
6. Schalow, B. J., Courcelle, C. T. & Courcelle, J. Mfd Is Required for Rapid Recovery of Transcription following UV-Induced DNA Damage but Not Oxidative DNA Damage in Escherichia coli. *J. Bacteriol.* **194**, 2637 (2012).
7. Ragheb, M. N. *et al.* Inhibiting the Evolution of Antibiotic Resistance. *Molecular Cell* **73**, 157-165.e5 (2019).
8. Dworsky P. Unfolding of the chromosome of Escherichia coli after treatment with rifampicin. *Z Allg Mikrobiol.* **15**, 243–7 (1975).
9. Ho, H. N., van Oijen, A. M. & Ghodke, H. The transcription-repair coupling factor Mfd associates with RNA polymerase in the absence of exogenous damage. *Nature Communications* **9**, 1570 (2018).
10. Le, T. T. *et al.* Mfd Dynamically Regulates Transcription via a Release and Catch-Up Mechanism. *Cell* **172**, 344-357.e15 (2018).

REVIEWER #3

The experiments are convincing that active transcription is important for NER in both the template strand and non-template strand. I am convinced as well that Mfd is not playing a crucial role in TCR. The experiments also do show that Mfd and Rho are somehow modulating TCR based transcription level of the gene. However, despite the authors' proposed model (which is pretty good), how this occurs is unclear and will require further exploration. Nevertheless, this is a strong paper and seems appropriate for publication in Nat Comm. I have

some minor comments that I think will strengthen the paper:

We thank the reviewer for their positive and constructive feedback. We do agree that the exact mechanism of our model will require experiments beyond the data shown in this manuscript. However, we would like to make the reviewer aware that this manuscript was first submitted as a co-submission with another manuscript (**Bharati et al.**) from our lab that compliments the data in this paper and provides more mechanistic studies of the pre-TCRC/TCRC model that is introduced here. Bharati et al used structural, biochemical, and genetic approaches to map the precise interactions RNAP makes with NER proteins to form the TCRC in vitro and in vivo. **We have uploaded this manuscript as “Related Manuscript File”** so that the reviewer can see the additional mechanistical data supporting the new TC-NER model we propose here.

Active Transcription is Required for GGR, Figure 1; The literature discussed in the introduction does not conclusively demonstrate that Mfd is not important for NER processes. Despite the strong data in the paper, I think it is premature to exclude Mfd from the experiments described in the earlier parts of the paper, especially given its previously reported roles in recovery from transcription stalling stress (e.g., UV irradiation). The discussion of Mfd later in the paper is fine, but in the first set of experiments, I find its omission problematic.

We agree that Mfd cannot be ruled out as a factor that influences NER and we do include experiments focused on Mfd in our manuscript. However, our paper is structured to first explain the indispensable role the ongoing transcription plays in NER. We then conclude our paper with an exploration of the role Mfd has in our updated model of TCR. We believe that including Mfd in the first figure would disrupt the logic of the paper and distract the readers from the main point of this figure- that transcription is necessary for the repair of CPD lesions in both the TS and NTS. Mfd is not a central element in our manuscript, however we do agree it is important to explore its role in repair and therefore, we devoted one section to this topic in our manuscript where Mfd vs WT +/- rif data is shown to the readers. **Please also see the additional (non-genomic) Mfd-related data and discussion in the co-submitted manuscript (Bharati et al).**

The rifampicin treatment used to achieve a true “transcription off” condition seems particularly stringent—while experimentally necessary and controlled for in downstream analyses, I wonder if this treatment in combination with UV may be excessive. Please describe.

The reviewer is right to wonder if these conditions are excessive. In fact, when we were planning the Rif experiment, we made sure to have necessary controls that would account for the stringent Rif conditions (such as the Western blots before and after Rif and ruling out that the repair deficiency wasn't due to a lack of NER enzymes and SOS induction). However, it is difficult to achieve high intracellular concentrations of Rif in gram(-) bacteria, making it bacteriostatic rather than bactericidal in *E. coli*. Additionally, an RNA-seq experiment from our lab (produced for an unrelated manuscript in preparation) shows that a “basal” level of transcription still occurs with Rif doses less than 100ug/mL. For example, a relatively low amount of the drug (50 µg/ml) is enough to stop *E. coli* growth, but not enough to stop all the transcription. To illustrate the difference between “high” (750 µg/ml) and “low” (50 µg/ml) Rif on transcription and NER, we performed the following experiments (**Extended Data Fig. 9 in Bharati et al.**). Using RT-qPCR we show that there is an approximately 100-fold difference in “residual” transcription at a representative highly active gene between cells treated with high and low Rif. Accordingly, quantitative CPD immunostaining shows that in contrast to high Rif, which prevented virtually any repair during 40 min of recovery from UV, low Rif still allowed much of the repair to occur.

The high Rif conditions we used would allow for a temporary halt in all ongoing transcription because it would allow enough intracellular Rif to be present. Because cells are not dying and we observed that all NER factors were present in our high Rif conditions, we were confident that the cells had the necessary components to repair CPD lesions, except for active transcription. In fact, Fig. 1c does show that $\Delta uvrA$ cells have decreased repair compared to WT +Rif cells, showing that our Rif conditions are not as harsh as a complete deletion in a core NER factor. These results may mean that a small amount of GGR occurs in absence of transcription. However, it is more likely that the Rif conditions we use are still not enough to completely abolish transcription or that the cell death that occurs in a $\Delta uvrA$ strain may make the overall repair look lower in that mutant. Regardless of the exact reason, in our complimentary manuscript (Bharati *et al.*), experiments are performed in specially designed transcriptionally “*insulated*” genomic regions, where we are sure there are no elongating RNAPs present, and we find that no NER can occur in this region without transcription.

*In Fig. 1c, “boxplots show the distribution of the percent decrease in TT-CPDs for each strand in gene bodies that had at least 1.5-fold TT-CPD enrichment at the 0-timepoint over NT-timepoint.” Please justify the 1.5-fold enrichment cutoff. This is repeated multiple times throughout the paper.

We thank the reviewer for noticing we did not include a justification for this important cutoff in the analysis. The 1.5-fold threshold was used to ensure that genes in the 0-timepoint had an enrichment for CPD-lesions of the NT-timepoint. Genes that did not show an increase in lesions after UV would show noise rather than actual repair over time and we therefore wanted to filter them out. **A distribution of the fold changes in the 0-timepoint over NT timepoint are now provided in Extended Data Figure 1 and show that the majority of genes pass this filtering step.**

Figure 2; Please elaborate on how high, medium, and low transcription levels were assigned, or where this information was originally published. Are the bins contiguous, or are subset of representative genes used? This will affect the interpretation of the results.

This is indeed an important information that we inadvertently left out. **We have addressed this by including how genes were split by transcription level in the methods section.** The genes were split based on RPKM levels from our RNA-seq data. The RPKM of each gene was used to split the genes into high- RPKM >30, mid – RPKM >5 and RPKM <=30 and low – RPKM <=5, transcription levels.

Fig. 2b is confusing. Upon close inspection, the trends the authors intend to illustrate are present, but it is difficult to critically analyze the data when figure contains so many different pieces.

We agree with this comment and **have made edits to Fig. 2b to try to make it clearer.** We have added a box around genes with an antisense transcription preference that were found in this analysis. We also added more details in the figure legend to make it clear what the figure is showing.

*NTS and Intergenic Regions are Subject to TCR; To determine which genes have an “antisense transcription preference,” the authors select a discrete region “in the last 100 bp and 50 bp downstream into intergenic regions.” I don’t have a major concern about these parameters, but because the group differences for the termination experiments are small, I

would be interested to see if the effect is altered by expanding/ relaxing the cutoffs here. A justification of the region would also be appropriate.

We have added Extended Data Figure 3 to our manuscript which shows this same analysis at three different windows in addition to the one shown in the main text. They all show an increase in genes with an antisense transcription preference compared to the entire gene body analysis as well as showing greater repair at the NTS gene end. We chose this cutoff based on a published study of Rho dependent termination sites¹. Although this analysis does not specifically analyze Rho terminators, the window used in this study provided a good starting point for our gene end analysis.

Figure 3; Fig. 3b is poorly designed and does not clearly illustrate the points made in the manuscript. The authors should make their point in a different way. Additionally, please keep a consistent color scheme between all the figures; the red and orange in these plots is inconsistent.

As requested, we have made several changes to Fig. 3b in order to emphasize the point we are trying to make in our manuscript. We do believe that a scatterplot is the best way to show this data because we can observe how the NTS in each gene changed after a mutation in Rho. These changes are also mild, so a scatterplot can clearly show the global, but mild increase in antisense transcription. However, we do agree that our original figure may have not done our results justice. We have therefore changed the color scheme and made each gene (represented by a point on the plot) easier to visualize by “zooming in” and reducing the range of both the x-axis and y-axis. We also added text to the scatter plot to help make the figure clearer. In addition, we updated the description in the figure legend so that it is easier for a reader to understand what the figure is showing. **We have made these changes for all of the scatter plots we use in our manuscript.**

Figure 4; In Figs. 4c, 4d, and 4e, a set of three genes is used to generalize the authors’ conclusions about UV-induced antitermination at Rho-dependent terminators. Please justify the selection of genes. The three included show consistent patterns with and without UV treatment but given the differences in magnitude in treatment effect I would like to see a few other terminators in the extended data.

These genes were chosen from a previously published study mapping Rho termination sites¹. We chose genes from this study that had a high dependency on Rho for termination. **We have also added Extended Data Fig. 7 where we show two additional Rho dependent termination sites.** We find that these terminators also have an increase in readthrough after UV exposure.

*Conclusions from each of these experiments would be strengthened with statistics. Absence of this analysis is particularly noticeable in the boxplots used repeatedly. With few exceptions, while the data do appear to show a trend, it is difficult to determine visually if this trend is significant.

We agree and have added statistics to all of our boxplot figures.

REVIEWERS' COMMENTS

Reviewer #1 (Remarks to the Author):

The authors have done a thoughtful thorough job of addressing the concerns I had raised and those raised by Reviewer 3. I am satisfied and continue to recommend publication of this manuscript as it will make a very important, highly novel contribution to the scientific community's understanding of these complex biological phenomena.

I was disturbed by the negative, almost destructive tone of Reviewer 2's enormously detailed comments, which seem to have the goal of preventing the publication of a thoughtful novel data-rich paper that challenges the assertion that Mfd is both necessary and sufficient for transcription-coupled repair. Blocking the publication of competing models is not the way that science progresses. The substantial body of published data indicating that the relationship between transcription and NER is more complicated than simply Mfd cannot be summarily dismissed by asserting that all these observations were shown to be incorrect by the 2017 Adebali et al. papers, which rely on the interpretation of TS/NTS ratios obtained using the XR-seq technique. Reviewer 2 repeatedly suggests that the Martinez et al. have failed to understand important results or principles, yet seems to misunderstand multiple issues themselves. For example, Reviewer 2 incorrectly repeatedly refers to the authors' high-resolution method as being "low-resolution" despite an independent publication about this method.

Reviewer #3 (Remarks to the Author):

The authors have thoroughly addressed reviewer comments and now present a strong manuscript that I confidently recommend for publication. The additional details included both in the text body and methods sections regarding statistical analysis and quantitative definitions of criteria used in computational models (e.g. low, medium, or high transcription levels) greatly increase the rigor of the work. Considering this work together with the experiments presented in the co-submitted manuscript, the authors present a compelling new model for bacterial transcription-coupled repair.